# Unraveling the Molecular Tumor-Promoting Regulation of Cofilin-1 in Pancreatic Cancer

**DOI:** 10.3390/cancers13040725

**Published:** 2021-02-10

**Authors:** Silke D. Werle, Julian D. Schwab, Marina Tatura, Sandra Kirchhoff, Robin Szekely, Ramona Diels, Nensi Ikonomi, Bence Sipos, Jan Sperveslage, Thomas M. Gress, Malte Buchholz, Hans A. Kestler

**Affiliations:** 1Institute of Medical Systems Biology, Ulm University, 89081 Ulm, Germany; silke.werle@web.de (S.D.W.); julian.schwab@uni-ulm.de (J.D.S.); robin.szekely@uni-ulm.de (R.S.); nensi.ikonomi@uni-ulm.de (N.I.); 2Department of Gastroenterology, Endocrinology and Metabolism, Philipps-University Marburg, 35043 Marburg, Germany; marina.tatura@staff.uni-marburg.de (M.T.); mel.sandra@gmx.de (S.K.); ramona.diels@staff.uni-marburg.de (R.D.); gress@med.uni-marburg.de (T.M.G.); malte.buchholz@staff.uni-marburg.de (M.B.); 3Institute of Pathology, University of Tübingen, 72076 Tübingen, Germany; bence.sipos@med.uni-tuebingen.de (B.S.); jan.sperveslage@ukmuenster.de (J.S.)

**Keywords:** pancreatic cancer, cofilin-1, modeling, molecular mechanism, Boolean networks, predicting therapeutic targets

## Abstract

**Simple Summary:**

Unraveling the mechanistic regulations that influence tumor behavior is an important step towards treatment. However, in vitro studies capture only small parts of the complex signaling cascades leading to tumor development. Mechanistic modeling, instead, allows a more holistic view of complex signaling pathways and their crosstalk. These models are able to suggest mechanistic regulations that can be validated by targeted and thus more cost-effective experiments. This article presents a logical model of pancreatic cancer cells with high cofilin-1 expression. The model includes migratory, proliferative, and apoptotic pathways as well as their crosstalk. Based on this model, mechanistic regulations affecting tumor promotion could be unraveled. Moreover, it was applied to screen for new therapeutic targets. The development of resistance mechanisms is a common limitation of cancer therapies. Therefore, new approaches are needed to identify optimal treatments. One is suggested in this article, indicating the surface protein CD44 as a promising target.

**Abstract:**

Cofilin-1 (CFL1) overexpression in pancreatic cancer correlates with high invasiveness and shorter survival. Besides a well-documented role in actin remodeling, additional cellular functions of CFL1 remain poorly understood. Here, we unraveled molecular tumor-promoting functions of CFL1 in pancreatic cancer. For this purpose, we first show that a knockdown of CFL1 results in reduced growth and proliferation rates in vitro and in vivo, while apoptosis is not induced. By mechanistic modeling we were able to predict the underlying regulation. Model simulations indicate that an imbalance in actin remodeling induces overexpression and activation of CFL1 by acting on transcription factor 7-like 2 (TCF7L2) and aurora kinase A (AURKA). Moreover, we could predict that CFL1 impacts proliferation and apoptosis via the signal transducer and activator of transcription 3 (STAT3). These initial model-based regulations could be substantiated by studying protein levels in pancreatic cancer cell lines and human datasets. Finally, we identified the surface protein CD44 as a promising therapeutic target for pancreatic cancer patients with high CFL1 expression.

## 1. Introduction

Pancreatic cancer is one of the most lethal cancers in developed countries [1,2]. With an estimated incidence of 458,918 new cases in 2018, it is the eleventh most common cancer in the world [1]. Since its mortality rate (432,242 cases [1]) nearly equals the incidence rate, pancreatic cancer is the fourth leading cause of cancer-related deaths [1,3,4]. Currently, curative resection of early-stage tumors followed by adjuvant chemotherapy offers the only hope of significant improvement of patient survival [5,6]. However, despite recent advances in therapy, the 5-year survival rates are still very low with approximately 5% [2,6]. Reasons are, among others, late diagnosis due to non-specific symptoms and remarkable resistance to chemotherapeutic treatment [3]. Thus, it is essential to uncover molecular mechanisms associated with the initiation and maintenance of the aggressive malignant phenotype of pancreatic cancer cells in order to identify potential targets for novel interventions.

High-throughput analyses and screenings of pancreatic tissues suggest cofilin-1 (CFL1) as a biomarker for pancreatic cancer [7,8,9,10]. In line with this, increased expression of CFL1 could be associated with proliferation, migration, or invasion in various types of cancer [11,12,13,14,15,16] as well as with a poorer overall survival [10,15,17,18]. The role of CFL1 in actin remodeling and thus migration and tissue invasion of cancer cells is well described [12,19]. However, still little is known about its role in regulating cell proliferation. Recently, Wang et al. [11] were able to show that inhibition of CFL1 in bladder cancer leads to cell cycle arrest in the G1-phase and an increase of the apoptosis rate. Taken together, these results encourage a deepened investigation of CFL1 involved pathways with a special interest in new therapeutic interventions. Therefore, we first performed RNAi-based functional experiments with CFL1 in pancreatic cancer cells, which later served as a basis for a gene regulatory network model to unravel mechanistic regulations.

## 2. Materials and Methods

### 2.1. Cell Lines

Panc-1 cells [20] were obtained from the American Type Culture Collection (Manassas, VA, USA). S2-007 as well as S2-028 were originally isolated from a liver metastasis of a PDAC patient [21] and were obtained from Takeshi Iwamura (Miyazaki Medical College, Miyazaki, Japan). IMIM-PC1 cells were also originally isolated from a liver metastasis [22] and were kindly provided by Francisco X Real (CNIO, Madrid, Spain). All cells were cultured in Dulbecco’s modified eagle medium (DMEM) containing 10% fetal bovine serum (FBS) and 0.05 mg/mL Gentamicin at 37 °C and 5% CO_2_ (*v/v*). We used the same cells and culture condition as previously mentioned [23].

### 2.2. siRNA Transfection

Pre-designed silencer siRNAs #146652 (5′-GGGAUCAAGCAUGAAUUGCtt-3′), #5455 (5′-GGAGAGCAAGAAGGAGGAUtt-3′) and #5366 (5′-GGACAAGAAGAACAUCAUCtt-3′) (Applied Biosystems/Ambion, Austin, TX, USA) were transfected in a concentration of 40 nmol using the siLentFect Lipid Reagent (Bio-Rad, München, Germany) according to manufacturer’s protocol. A cell non-targeting siRNA (Thermo Fisher Scientific, Dharmacon Products, Lafayette, LA, USA) was employed as control.

### 2.3. Stable Repression of CFL1 Expression

For inducible repression of CFL1, a CFL1-specific shRNA construct (forward: 5′-ccg gga gga caa gaa gaa cat cat cct cga gga tga tgt tct tct tgt cct ctt ttt g-3′, reverse: 5′-aat tca aaa aga gga caa gaa gaa cat cat cct cga gga tga tgt tct tct tgt cct c-3′) was cloned into the tetracycline-inducible pLKO-U6-Tetr-on vector backbone (kindly provided by Dr. Stephan Hahn, Dept. of Gastroenterology, University of Bochum, Bochum, Germany). Plasmid DNA was extracted and transfected into S2-007 cells. Stably transfected cell clones were selected by adding puromycin to a concentration of 1.4 µg/mL.

### 2.4. In Vivo Xenograft Assays

2 × 10^6^ S2-007 cells in 0.1 mL PBS were injected subcutaneously into the right flank of 6 weeks old female nude mice. For induction of shRNA expression, doxycycline was added to the drinking water of the mice (2 mg/mL + 2% sucrose); control groups received normal water. Unspecific effects of sucrose addition on tumor growth were excluded in previous studies using the same experimental setup [24,25]. Health status of the animals, including possible signs of dehydration, was controlled daily. Six animals per arm, in total 12 were implanted all developing tumors. Tumor sizes were measured twice weekly using a caliper. Upon sacrificing the mice, resected tumors were frozen in liquid nitrogen and stored for RNA and protein analyses. Animal experiments were approved by the relevant Ethics Committee at the Regierungspräsidium Giessen, Germany (ethic approval number V54 19c 20 15 (1) MR 20/11 Nr. 50/2011).

### 2.5. qRT-PCR

RNA was isolated according to the manufacturer’s protocol with the peqGold Total RNA Kit (peqlab Biotechnology GmbH, Erlangen, Germany). NanoDrop ND-1000 (peqlab Biotechnology GmbH) was used to determine the RNA concentration. First-strand cDNA synthesis was performed with 1 µg total RNA and the Omniscript Kit (Qiagen, Hilden, Germany) according to the manufacturer’s instructions. cDNA amplification was carried out in 7500 Fast Real-Time PCR System (Applied Biosystems, Warrington, FL, USA) with the Power SYBR Green Master Mix (Applied Biosystems) and specific primer pairs. The primers CFL1 (CFL1 forward: 5′-CCTATGAGACCAAGGAGAGCAAGA-3′ reverse: 5′-TTGGAGCTGGCATAAATCATTTT-3′) and ribosomal protein large P0 (RPLP0, forward: 5′-AGTTTCTCCAGAGCTGGGTTGT-3′ reverse: 5′-TGGGCAAGAACACCATGATG-3′) were designed using the Primer Express program (Applied Biosystems).

### 2.6. Western Blot

Proteins were isolated by centrifugation of cells in culture medium at 4 °C for 5 min and 16,000× *g*. If total amount of proteins was extracted, pellets were resuspended in proteinase inhibitor (G-Biosciences, Maryland Heights, MO, USA) supplemented PBS before sonicated with a Labsonic U (B.Braun, Melsungen, Germany). Instead, cell fractions were obtained by resuspending initial centrifugated cells again in ice-cold PBS before centrifugation for another round of 3 min at 6000× *g* and 4 °C. The obtained pellet for the nuclear fraction was then resolved in 200 µL of proteinase inhibitor supplemented fractionation buffer (20 mM HEPES-KOH, pH 7.5, 10 mM KCl, 1 mM EGTA, 1 mM DTT, 0.5 mM PMSF, supplemented with proteinase inhibitor) and kept on ice for 30 min. Afterwards, it was squeezed five times through a 26G needle and centrifuged for 10 min at 1000× *g*. Next, supernatant and pellet were separated. The separated pellet was resuspended again in 50 µL fraction buffer and designated as nuclear fraction. Instead, the separated supernatant was centrifugated for 10 min at 16,000× *g* and 4 °C before designating it as cytosolic fraction. Bradford assay measured with the Multiskan FC photometer (Thermo Scientific, Langenselbold, Germany) was used to determine protein concentrations. For the SDS-PAGE, 10 µg proteins were loaded on 10 or 15% gel in SDS buffer and separated at 120V voltage. Afterwards, proteins were transferred onto a nitrocellulose membrane (Whatman GmbH, Dassel, Germany) at 300 mA current using a semi-dry blotting system (Biozym, Hessisch Oldendorf, Germany), and blocked for 4–6 h at 4 °C in 1×TBS, 0.1% Tween 20 (TBS-T) and 5% powdered milk. Primary antibodies (anti-CFL1 #ab42824, Abcam, Cambridge, UK; anti-Caspase-3 #9662, Cell Signaling Technology, Danvers, MA, USA; anti-Cleaved Caspase-3 #9661, Cell Signaling Technology; anti-PARP #9542, Cell Signaling Technology; anti-STAT3 #9139, Cell Signaling Technology; anti-phospho-STAT3 (Tyr705) #9145, Cell Signaling Technology; anti-Cyclin D1 #ab16663, Abcam; and anti-Myc #sc-481, Santa Cruz Biotechnology, Santa Cruz, CA, USA) were diluted 1:1,000 in blocking buffer and incubated overnight at 4 °C and washed afterwards in 0.1% TBS-T. To ensure equal loading, actin (anti-β-actin #sc-1616 Santa Cruz Biotechnology) or Lamin A/C (Lamin A/C #2032, Cell Signaling Technology) were used. The secondary antibody (anti-rabbit #7074, Cell Signaling Technology) was 1: 10,000 diluted in blocking buffer and incubated for 1–2 h at 4 °C. Proteins were detected using the ECL immunoblot kit (GE Healthcare Europe GmbH, Freiburg, Germany). Quantification of CFL1 bands in the western blots have been performed by densiometric analyses via Image-J. Band intensity was normalized to β-actin for all samples.

### 2.7. BrdU Proliferation Assay

The bromodeoxyuridine (BrdU) proliferation assay was performed using the Cell Proliferation ELISA Kit (Roche Diagnostics, Mannheim, Germany). Therefore, 24 h post-transfection, appropriate number of cells (Panc1: 7000 cells; S2-007: 5000 cells; S2-028: 7000 cells; IMIM-PC1: 7500 cells) were reseeded in a ViewPlate-96 Black cell culture plate (Perkin Elmer, Waltham, MA, USA). Following over-night culture, cells were incubated in BrdU containing medium for 4 to 6 h. After fixation for 1 h, cells were stained with anti-BrdU antibody for 1.5 h and chemiluminescence substrate was added to emit light. Emitted light was quantified with a Centro LB 960 luminometer (Berthold Technologies GmbH, Stuttgart, Germany).

### 2.8. Soft Agar Assays

24 h post-transfection, 5000 S2-007 cells or 7000 Panc-1 cells were mixed with DMEM containing 0.33 % Bacto Agar (Becton, Dickinson & Company, Sparks, NV, USA) and reseeded in 12-well cell culture plates coated with a bottom layer of 0.5 % Bacto Agar in DMEM. Viable colonies were counted 7–10 days after reseeding under a microscope with 10× magnification.

### 2.9. Flow Cytometry

Cells were trypsinized and centrifuged for 3 min at 1000× *g*. Following washing with PBS, the pellet was resuspended in 50–100 µL PBS and the suspension was transferred in a drop-wise manner into ice-cold ethanol (70%) under continuous vortexing. After centrifugation for 5 min at 1000× *g* and discarding of the supernatant, the remaining pellet was washed with 500 µL ice-cold PBS and resuspended in a propidium-iodide mixture (1 mg/mL Propidium-iodide (Sigma Aldrich, St. Louis, MO, USA) + 500 µg/mL DNAse-free RNAse (Roche Diagnostics, Mannheim, Germany) in PBS). Following incubation in darkness for 30 to 45 min at room temperature (RT), the sample was measured subsequently with the flow cytometer LSR II (BD Biosciences, Heidelberg, Germany). Recorded data were evaluated using the software ModFit LT (Verity Software House, Topsham, ME, USA).

### 2.10. Cell Tracking

To measure undirected cell migration, S2-007 cells (20,000 cells per well) or Panc-1 cells (30,000 cells per well) were seeded in collagen-coated 6-well culture plates and recorded under the microscope. We used a microscope from the Zeiss Cell Observer system (Carl Zeiss GmbH, Jena, Germany) with temperature as well as CO_2_ control. Pictures were recorded in 10 min intervals. The resulting time-lapse video files were analyzed with the Time Lapse Analyzer software [26] by extracting paths for each cell and calculating the average velocity of migration in µm/min.

### 2.11. Wound Healing

A 200 µL Diamond Tip (Gilson Inc., Middleton, WI, USA) was used to scratch a wound of 1–2 mm into a confluent layer of transfected cells in a 6-well plate and the medium replaced with fresh complete medium (10% FBS). Cells were seeded at different densities to achieve confluence within similar timelines. Pictures of regions of interest were recorded in intervals of 10 min and the resulting time-lapse video files analyzed using the Time Lapse Analyzer software [26]. Wound closure rates were expressed as a decrease in wound area over the duration of the experiment (µm^2^/h).

### 2.12. Boolean Network

Mechanistic models are inferred from experimental observations. However, many of these mechanistic models require kinetic parameters thus limiting their use. Logical models instead can be conceptualized on qualitative knowledge. Here, regulatory interactions are formulized by logical connectives. Boolean network models are a popular modeling paradigm among these gene regulatory networks [27,28,29,30,31,32]. The simplicity of this model arises from the assumption that genes and proteins are considered as either expressed/active or not expressed/inactive and regulatory interactions are formulized by logical connectives [33,34]. Information regarding regulatory interactions is derived from qualitative literature statements summarizing information from multiple sources e.g., publications (in vitro and in vivo studies), databases, or clinical trials. This causes the process of modeling to be iterative to ensure that any enlargement of the model still recapitulates the expected behavior. While molecular and biochemical experiments are preferable to build logical connections, validation of long-term behavior is made through phenotypical studies (e.g., proliferation assays, apoptosis measures, or cell cycle analyses). This validation is possible only by studying the dynamic behavior of the established model. In particular, steady states so called attractors, describe the long-term behavior of the model and have been associated to biological phenotypes [33,35]. These attractors can be either single state ones or a cyclic sequence of states. Studying long-term behavior together with paths leading to attractors (steady states) allow to unravel mechanistic regulations [33] and cell fates [35] of different weights [36,37]. Mechanistic models also allow to study perturbation of dynamic behavior. These in silico experiments on the mathematical model are similar to in vivo or in vitro knock out or knock in experiments on model organisms. For this reason, they are of crucial interest in predicting intervention targets and further guiding future laboratory experiments [37,38]. Altogether, this approach allows to reduce both time and costs for experimental setups by suggesting mechanisms and intervention targets.

### 2.13. Model Construction

For the presented model, regulatory interactions of network components were manually extracted from literature. In general, we started summarizing reviews on PDAC and molecular features of CFL1 overexpression. Whenever possible, information from PDAC context was considered. Here we integrated studies from various mouse models and pancreatic cancer cell lines. When possible, human studies were also considered with special focus on CFL1 expressing tumors. Moreover, data providing evidence for direct interactions (phosphorylation, binding, transcription) were weighted with major priority. In further refinements, data from indirect interactions was considered (expression correlations).

The aim of the presented model is to investigate mechanisms involved in CFL1 overexpression in PDACs and its role in cancer progression. In particular, a special focus of the model was to unravel the tumor promoting function of CFL1 with respect to cell survival and proliferation. This information was integrated with CFL1’s most studied role in cellular motility. Based on our initial in vitro and in vivo studies, we started the model setup by integrating regulators of CFL1, the cell cycle, as well as apoptosis inducers. For this purpose, we applied the search terms “CFL1 pancreatic cancer”, “CFL1 regulation”, “CFL1 cell cycle”, or “apoptosis pancreatic cancer” in Google. Afterward, we focused on studies describing CFL1 interventions and the described effect on other proteins (on expression or activity levels). In this perspective, our Google search terms were “CFL1 knockout” or “CFL1 overexpression”. Finally, the identified proteins within the model were analyzed concerning their mutual regulation. All identified regulations were summarized by logical connectives into a mechanistical model. Please note again, that throughout the whole modeling process, studies or reviews of PDAC were preferred. Only in cases where nothing related to PDAC could be found or to confirm a link, information from other cancers was considered.

### 2.14. Model Simulation

Attractor search and knockout simulations were performed using the R-package BoolNet [39,40]. Here, we utilized an exhaustive attractor search with a synchronous updating scheme. For the estimation of the basin of attraction of the attractors from the intervention screening performed with a SAT solver we also performed an exhaustive attractor search with limited to 1,000,000 initial start states.

### 2.15. Perturbation Screening

Systematic perturbation screenings were performed with the Java-based framework ViSiBooL [41,42] taking account of up to two combinations of interventions and that all proteins/genes in the network can change except caspases (Appendix A). Based on the established Boolean network of CFL1 in PDAC a requirement to induce apoptosis is the activity of caspases. Thus, automatic screening was performed to search for combinations that active caspases in the attractor.

### 2.16. Binarization of Gene Expression Data

We considered three microarray studies comparing normal pancreatic tissues with pancreatic cancer tissues (GSE15471, GSE32676, GSE16515). Please note, that the dataset GSE15471 contains matched tissues taken at the time of resection. All these microarrays were performed on Affymetrix U133 Plus 2.0 whole-genome chips. Transcriptional regulated proteins that were included in the model were mapped according to their Entrez ID to the hgu133plus2 SYMBOLs. Their expression data was robust multi-array average (RMA) normalized [43] and binarized via a threshold defined by a ROC curve using the R-package pROC [44] (Appendix A). According to this threshold expression values above the threshold were considered as active and expression values below the threshold as inactive. Three samples in the GSE15471 dataset contain replicates. These replicates were averaged and considered as a single data point.

## 3. Results

### 3.1. CFL1 Is Overexpressed in Pancreatic Cancer

As a first step in the depth-characterization of CFL1 in pancreatic cancer, we analyzed its expression in primary human pancreatic cancer and control tissues. Quantitative RealTime PCR analyses demonstrated strong significant overexpression of CFL1 mRNA in pancreatic tissue (*n* = 12) both in comparison to healthy pancreas tissue (*n* = 8) from organ donors (*p* = 0.0005) as well as to chronic pancreatitis (C.P., *n* = 9) cases (*p* = 0.00008), while expression in chronic pancreatitis and healthy pancreas were not significantly different (*p* = 0.54) (Figure 1a). As expected, CFL1 expression was also high in all pancreatic cancer cell lines examined. To examine the functional role of CFL1 in pancreatic cancer cells, knockdown of endogenous CFL1 with three independent siRNAs was performed. As normal CFL1 expressing non-tumoral control, we used the well-characterized HEK-293 cell line. Thereby we achieved a CFL1 knockdown efficiency of 60–90% on the RNA level (Appendix A) which was likewise reflected on the protein level (Figure 1b, and Appendix A).

### 3.2. CFL1 Knockdown Inhibits Proliferation and Tumor Growth

BrdU incorporation assays revealed inhibition of proliferation in CFL1 silenced pancreatic cancer cell lines cancer cell lines (Figure 1c). In line with this observation, a decrease in colony formation and size was observed after treatment with siRNA against CFL1 (Appendix A). Next, we studied the cell cycle phase impacted by CFL1 silencing with flow cytometric analyses (Figure 1d). Similar to the findings of Wang et al. [11] in bladder cancer, we observed a reduced number of actively proliferating cells (S-phase) accompanied by an increase of cells in the G0/G1-phase indicating an attenuation of the G1/S-phase transition.

To assess the importance of CFL1 expression for the growth of pancreatic cancer cells in vivo, xenograft tumors were induced in nude mice by subcutaneous injection of S2-007 cells stably transfected with an inducible CFL1 shRNA. Induction of shRNA CFL1 expression led to a significant reduction of CFL1 on mRNA as well as on the protein level (Figure 1e and Appendix A). Moreover, CFL1 repression resulted in reduced tumor volume (Figure 1f). These results demonstrated that CFL1 has a growth-supporting role in pancreatic cancer.

### 3.3. CFL1 Knockdown Does Not Induce Apoptosis

Studies in bladder and vulvar squamous carcinoma describe an increase in the apoptosis rate after CFL1 knockdown [11,45]. In contrast to this, we did not observe induction of apoptosis after silencing of CFL1 in pancreatic cancer cells (Figure 2a and Appendix A). None of the pancreatic cancer cell lines treated with CFL1 siRNA showed cleaved-caspase 3 or cleaved PARP activity–both indications for apoptosis - nor did any of the cells treated with non-silencing siRNA or untreated cells. These results indicate distinct differences in the response of pancreatic cancer cells to CFL1 silencing, and by extension distinct differences in CFL1-associated signaling networks, compared to other types of cancers.

### 3.4. CFL1 Deficiency Leads to Distinct Defects in Cell Migration

Since CFL1 has previously been described to influence the migratory potential of cancer cells, we measured the influence of CFL1 knockdown on the migratory potential of cells by cell tracking and wound healing experiments (Figure 2b,c, and Appendix A). Here, we observed reduced cell velocity as well as a decreased potential for wound closing. The effect was more pronounced for S2-007 cells compared to Panc-1 cells, which may be explained by the fact that the liver metastasis-derived S2-007 cells show a more aggressive growth behavior overall compared to Panc-1 cells, which originate from a primary tumor. Hence, similar to other cancers, CFL1 silencing decreases infiltration and migration potential of pancreatic cancer cells.

### 3.5. Establishing a Model to Uncover Mechanistic Regulation of CFL1

Besides its well-described role in actin remodeling, regulatory mechanisms inducing CFL1 overexpression or affecting apoptosis, and proliferation downstream of CFL1 are unknown. From the results above (Section 3.1, Section 3.2, Section 3.3 and Section 3.4), we were able to show that CFL1 silencing affects these processes. Nevertheless, these results provide only phenotypical descriptions. To uncover tumor-promoting mechanisms of CFL1, we built a gene regulatory network. This model was based on the functional data described above as well as an intensive literature search. Moreover, CFL1 regulators and proteins involved in migration, proliferation, and apoptosis were included. A detailed description of the considered regulatory interactions included in the final mechanistic model can be found in Table 1. The final model consisted of 33 nodes and 130 interactions capturing actin-remodeling, CFL1 regulation as well as cell cycle regulation and apoptosis induction (Figure 3a). Systematic evaluation of global network dynamics over time showed that this model is able to recapitulate our previous observations in pancreatic cancers as well as for CFL1 silenced cells (Figure 3b,c). The gene regulatory network shows in its stable state (attractor) active CFL1 in combination with a proliferative and infiltrating phenotype (active S-phase and F-actin_new_) but no induction of apoptosis (inactive caspases). In contrast, the analyses of CFL1 knockout revealed an attractor representing inactive proliferation and infiltration but still without active apoptosis. Thus, our newly established gene regulatory model might be suitable to uncover mechanistic regulations behind the dynamics leading to the more severe phenotype of active CFL1 in pancreatic cancer. To do so, we analyzed the network progression from healthy (unstimulated state) towards cancer. Since activating KRAS mutations can be found as one of the earliest mutations in approximately 90% of pancreatic cancer patients [46], we started from a state with active KRAS. Originated cascades (Figure 3b,c) were analyzed in detail to uncover cancer driving mechanisms. Model-based predicted activities of proteins were compared to literature or supported by laboratory experiments and human dataset analyses. While detailed analyses of the progression are described in the following paragraphs, a summary of this comparison can be found in Table 2.

### 3.6. Ras-Induced Imbalance in Actin Remodeling Leads to Overexpressed and Activated CFL1

First, we concentrated on processes leading to overexpression and activation of CFL1. Following the progression towards cancer, our model shows an imbalance between the two opponents ras homolog family member A (RHOA) and ras-related botulinum toxin substrate 1 (RAC1) in favor of RAC1 (Figure 3b). Thus, protein kinase D1 (PRKD1) downstream of RHOA is rendered inactive and enables the expression of TCF7L2, an inducer of CFL1 expression. On the other hand, by acting on p21 activating kinase 1 (PAK1), RAC1 activates aurora kinase A (AURKA). AURKA phosphorylates and thus activates slingshot-1L (SSH1L), one of the activators of CFL1. Based on the network evolution over time, we assume that an imbalance in actin remodeling induced by KRAS acting on PI3K results in overexpression and activation of CFL1.

Our mechanistic hypothesis is supported by gene expression data, showing that both AURKA and TCF7L2 are significantly overexpressed in pancreatic tumor tissues in comparison to healthy donors (Figure 4a,b). Here, binarization of the expression data classified tumor samples as active in contrast to healthy samples. As a further support, results were compared to literature findings (Table 2).

### 3.7. The Synergy between CFL1 and Arp2/3 Is Important for Migration

Pancreatic cancer cells are characterized by a high level of early and aggressive metastasis [3]. We have already been able to show that CFL1 influences this migratory potential (Figure 2). In general, it is assumed that non-phosphorylated CFL1 severs filamentous actin fibers (F-actin) preferentially at old adenosine diphosphate (ADP)-F-actin ends thus providing glomerulus actin (G-actin) for the generation of newly polymerized F-actin [58,68,77]. Although this mechanism is certainly applicable to a large number of cell types, some recent studies have shown that CFL1 is also involved in lamellipodia formation [58,68,71], which is the driving force of cancer cell migration [133,139].

Insights on the dynamic nature of this regulatory network support this assumption (Figure 3b,c). Here, actin-related protein 2/3 complex (ARP2/3) downstream of RAC1 is unable to synthesize new actin filaments unless CFL1 is activated. Based on this, we suggest that CFL1 and ARP2/3 work in synergy for cell spreading.

In literature, we found that this synergy might work through the severing capabilities of CLF1 providing short actin fibers that are preferentially used by ARP2/3 to create new branched actin fibers [134].

### 3.8. CFL1 Influences the Cell Cycle via STAT3

Next, we concentrated on signaling pathways describing how CFL1 might influence the cell cycle. Tsai et al. [140] suggest a cell cycle inhibitor dependent regulation. That would also be possible in pancreatic cancer if these inhibitors are excluded from the nucleus by enhanced AKT activity [109]. However, we could show that neither p21 and p27 protein levels nor their cellular localization changed in response to variations in CFL1 levels (Figure 4c and Appendix A).

In the presented gene regulatory network, the next protein being activated in the transition towards pancreatic cancer (Figure 3b) after CFL1 is STAT3 (time step 8). However, STAT3 alone is not able to promote cell cycle progression through activation of cyclin D1 (CCND1), because CCND1 is not activated before time step 12. After activation, STAT3 activates protein kinase B (AKT) further enabling β-catenin (CTNNB1) to induce MYC expression by inhibiting glycogen synthase 3β (GSK3B) (time step 9–11). Consequently, the MYC proto-oncogene induces the expression of CCND1. Due to already active AKT, CCND1 can associate with its cyclin-dependent kinase (considered together with CCND1) and translocate into the nucleus. Here, it phosphorylates retinoblastoma (RB), thereby freeing the E2F transcription factor (E2F) which further induces the expression of Cyclin E2 (CCNE1) and thus the progression from G1 to S-phase. Conversely, all these proteins stay inactive in the systemic CFL1 knockout simulation and also cyclins stay inactive. This model behavior can be understood as attenuation of G1- to S-phase transition (Figure 3c) which we could show to take place in CFL1 silenced pancreatic cancer cells (Figure 1d).

To confirm the model-based assumption that CFL1 regulates the cell cycle via STAT3, we investigated the impact of active CFL1 and an in-silico STAT3 knockout on the dynamic network behavior by further simulations (Appendix A). Although this simulation shows active CFL1 and indicates migratory potential of the cells (active F-actin_new_), the in-silico knockout of STAT3 inhibits proliferation (inactive cyclins). Contrary to CFL1 knockout simulation, however, in-silico STAT3 knockout was able to induce apoptosis. Based on these observations, we concluded that CFL1 influences the cell cycle via STAT3.

In order to support this theory, we checked protein levels of STAT3, MYC, and CCND1 in CFL1 knockdown and control cells by western blots (Figure 4d and Appendix A). Thereby we observed that both total STAT3 as well as active STAT3 protein levels decreased after CFL1 silencing. Similar results were found for MYC and CCND1 thus strongly supporting model simulations.

### 3.9. Mitochondrial CFL1 and Its Downstream Targets Influence Apoptosis Regulation

Pancreatic cancer is characterized by a high degree of apoptosis resistance. One reason for this is might be that pancreatic cancer cells require death-receptor signals as well as the mitochondrial enhancing signal to induce apoptosis [119,120].

Although our simulation indicates activation of cytochrome C (CYCS) and thus its release into the cytoplasm in the transition towards cancer (Figure 3b, time step 9), there is no induction of apoptosis as evidenced by inactive caspases. The simulation shows active STAT3 and AKT shortly before CYCS is activated. Both are known to influence apoptosis by inducing expression of anti-apoptotic factors or phosphorylation of caspases [119,120]. Based on these findings, we assume that apoptosis of pancreatic cancer cells is inhibited by unbalanced expression of anti-apoptotic proteins, and the activity of AKT.

To support this model-based hypothesis, we studied the expression of BCL2L1 in human gene expression datasets (Figure 4e). This anti-apoptotic protein is known to be regulated downstream of STAT3. We observed significant overexpression of BCL2L1 in pancreatic tumor tissues. Besides, binarization of the expression data classified BCL2L1 as active (Figure 4e).

Contrary to model analyses with active CFL1, AKT and STAT3 are both inactive in CFL1 knockout simulations (confirmed for STAT3 by molecular data see Figure 4d), and CYCS remains inactive throughout all time steps (Figure 3c). This simulation outcome points to an additional role of CFL1 in apoptosis induction.

Interestingly, we found no difference in BAX expression levels between samples from pancreatic tumors or normal controls (Figure 4f). Independent studies describe a translocation of activated CFL1 to the mitochondrion after apoptosis induction. In this context, CFL1 acts as a carrier for the pro-apoptotic BAX protein [122,126].

Combining our findings with previously described CFL1-dependent mechanisms, we postulate that although inactivation of AKT and STAT3 in response to CFL1 knockdown would normally lead to CYCS release and subsequent activation of caspases in pancreatic cancer cells, BAX activation is prevented by lack of availability of unphosphorylated CFL1, thus counteracting apoptosis induction in this case.

### 3.10. Systematic Perturbation Screening Identified Targets to Induce Apoptosis in the Model

Even though our mechanistic model necessarily represents an oversimplification of complex cellular networks, we could show that it is able to reproduce the behavior of CFL1 in pancreatic cancer. Consequently, we tested if we could apply it to screen for therapeutic targets that may trigger the induction of apoptosis in pancreatic cancer. Based on our model, this would be the case if caspases become active. Manually screening for promising intervention targets in larger networks is time-consuming, if not even impossible when considering combinatorial approaches [42]. Based on the presented CFL1 model, we had to test 2048 possible combinations to screen for perturbation of at least two proteins [42]. Thus, we used the Java-based framework ViSiBooL to screen systematically for promising intervention targets which uses SAT solvers for fast exhaustive attractor search.

A single target intervention screening based on the established model identified three proteins (CD44, STAT3, and TWIST1) which were predicted to induce apoptosis to 100% (Figure 5 and Appendix A). Besides, we searched for combinations of targets to induce apoptosis in the model. Here, the model-based perturbation screening proposed a list of 14 different combinations of interventions (Appendix A). Please note, that one limitation of the applied SAT algorithm is that it does not return state transitions. For this purpose, further detailed simulations with the previously suggested targets were performed. By taking into account the biological importance of the individual proteins and their dynamic impact on the final phenotype, we finally identified AURKA and PAK1 as further intervention candidates. According to the intervention screening, both induce apoptosis in combination with active CFL1 or active actin (which are both present in pancreatic cancer). Furthermore, simulations of a single intervention of AURKA or PAK1 lead in >99% to attractors representing apoptosis (Figure 5 and Appendix A).

## 4. Discussion

Various studies in different types of cancers describe a correlation of CFL1 expression with an aggressive phenotype and worse prognosis for patients. High-throughput screenings of pancreatic cancers already suggested CFL1 as biomarker [7,8,9,10]. Similar to Satoh et al. [10], we could show that CFL1 expression is specifically increased in pancreatic cancer but not in chronic pancreatitis (Figure 1) and that silencing of CFL1 is associated with a reduction of migration (Figure 2). However, our in-depth characterization of CFL1 in pancreatic cancer further supports a central and much more complex role of CFL1, including regulatory functions in proliferation and apoptosis inhibition. Although several of our applied pancreatic cancer cell lines were originally derived from liver metastases of pancreatic tumors, this does not invalidate their use as in vitro models of PDAC. In this perspective, a number of studies have established that metastases share the phenotypic traits of the primary tumor from which the derive [141,142]. Our selected cell lines represent a spectrum of different grades of differentiation and invasiveness of PDACs, thereby avoiding falsely identifying effects which are in reality artefacts in a single cell line.

Although an association of CFL1 overexpression with cancer progression has previously been established, the molecular interactions regulating its behavior in cancer are far from understood. Here, we used a mechanistic model to unravel molecular tumor-promoting regulations of CFL1 in pancreatic cancer. Systems biology is an interdisciplinary approach that studies signaling crosstalk holistically instead of studying small parts or single interactions within a signaling cascade. Thus, to the best of our knowledge, our model is the first that captures holistically the regulation of CFL1 and its impact on cancer progression.

Since existing knowledge about regulatory interactions in biology is mostly qualitative and kinetic parameters are often not available, gene regulatory network models are an appropriate tool to initially uncover pathway regulation and their crosstalk. Despite their simplicity, they are able to reproduce complex behavior and help to guide biological research. Likewise, the CFL1 model is capable of reproducing the previously observed functional and molecular events in pancreatic cancer cells with and without RNAi-mediated knockdown of CFL1 expression. Furthermore, model-based assumptions on regulations could be corroborated by additional in vitro experiments.

In biology, it is well known that several regulatory interactions are cell type specific. For instance, for some cell entities, the death receptor stimulation is sufficient to induce apoptosis. In contrast, pancreatic cancer cells belong to a group of cells that require an additional mitochondrial enhancing signal to induce apoptosis (“Type 2 cells”) [119]. This may explain why CFL1 knockdown already triggers apoptosis in other cancers [45] but not in pancreatic cancer (Figure 2a). Within our model, the lack of apoptosis induction is explained by a dependency of BAX on the availability of CFL1 for efficient translocation to the mitochondria. While CFL1 knockdown thus generates an initial pro-apoptotic stimulus, it simultaneously interrupts the apoptotic cascade on the level of BAX mitochondrial translocation.

The regulation of cell cycle progression by CFL1 shows similar cell-specific aspects. While Tsai et al. [140] suggested regulation of proliferation by attenuation of cell cycle inhibitors in human non-small lung cancer cells, this regulation can be excluded for pancreatic cancer cells. We found neither a change in the protein levels of the prominent cell cycle inhibitors p21 and p27 nor a change in their localization after CFL1 depletion (Figure 4c). Conversely, our simulations supported the conclusion of Wu et al. [45] that the effect of CFL1 on the cell cycle is mediated by STAT3.

An obvious advantage of having access to mathematical models that faithfully reproduce complex molecular interactions is the possibility to simulate pharmacological inhibition of single targets or combinations of targets within the network at practically no cost. Since the regulation of apoptosis is prominently featured in our model, this was an obvious choice as a “functional readout” for screening for potential intervention targets.

The model simulations proposed the proteins AURKA, CD44, PAK1, STAT3, and TWIST1 as promising therapeutic targets for pancreatic cancer. Interestingly, there are already ongoing clinical trials performed with several AURKA (NCT00249301, NCT01924260) and STAT3 (NCT02983578, NCT03382340) inhibitors with pancreatic cancer patients (https://clinicaltrials.gov (accessed on 5 January 2021)), further supporting the relevance of the model’s conclusions. However, while ongoing clinical trials are performed with several CD44 inhibitors (e.g., NCT02046928, NCT03078400), none is applied to pancreatic cancer patients. In contrast, no compounds are available yet which are suitable for PAK1 or TWIST1 inhibition in humans [87]. The most promising compound for PAK1 inhibition, PF-3758309, failed due to its low oral bioavailability in humans while other inhibitors like IPA-3 or G-555 reveal either cellular toxicity or cardiovascular toxicity respectively [63,87]. The same is true for the first TWIST1 inhibitor. Recently, Yochum et al. [143] published harmine as a TWIST1 inhibitor. However, while they did not observe toxicity in their in vivo model, harmine is associated with neurotoxicity in humans [144].

According to our model, CD44 may be a particularly attractive novel target in pancreatic cancer. This is further supported by in vitro and in vivo preclinical studies showing decreased migration and growth in pancreatic cancer cells after CD44 knockdown [145,146,147,148]. In this perspective, it should be highlighted that some of these experiments were performed in Panc-1 cell lines that have a high CFL1 expression (Appendix A). Moreover, it could be shown that a decrease in CD44 levels leads to a reduction in the activity of STAT3 and AKT [118,146], which is also replicated in our attractor (Appendix A). The simultaneous loss of AKT and STAT3 activity may prove particularly effective because both pathways are described to contribute to chemotherapy resistance [149] (as also described for CD44 [146,150]). However, the efficacy of PAK1, TWIST1, or CD44 inhibition for pancreatic cancer treatment will have to be determined in further pre-clinical and clinical studies.

Taken together, our results provide compelling evidence for an important, multi-faceted pro-oncogenic role of CFL1 in pancreatic cancer cells in vitro and in vivo.

## 5. Conclusions

We present a Boolean network model which accurately reflects functional and molecular observations of pancreatic cancer with high cofilin-1 (CFL1) expression. This mechanistic model is able to predict the behavior of CFL1 and its effect on downstream targets in pancreatic cancer cells. Analyses of dynamic behaviors allowed to hypothesize about molecular mechanisms of sustaining CFL1 overexpression, its impact on cell cycle, invasion, and apoptosis. Moreover, this model allows simulating pharmacological interventions in order to identify potential novel drug targets. Thereby, we identified CD44 as promising drug target for pancreatic cancer patients with high CFL1 expression.

## Figures and Tables

**Figure 1 cancers-13-00725-f001:**
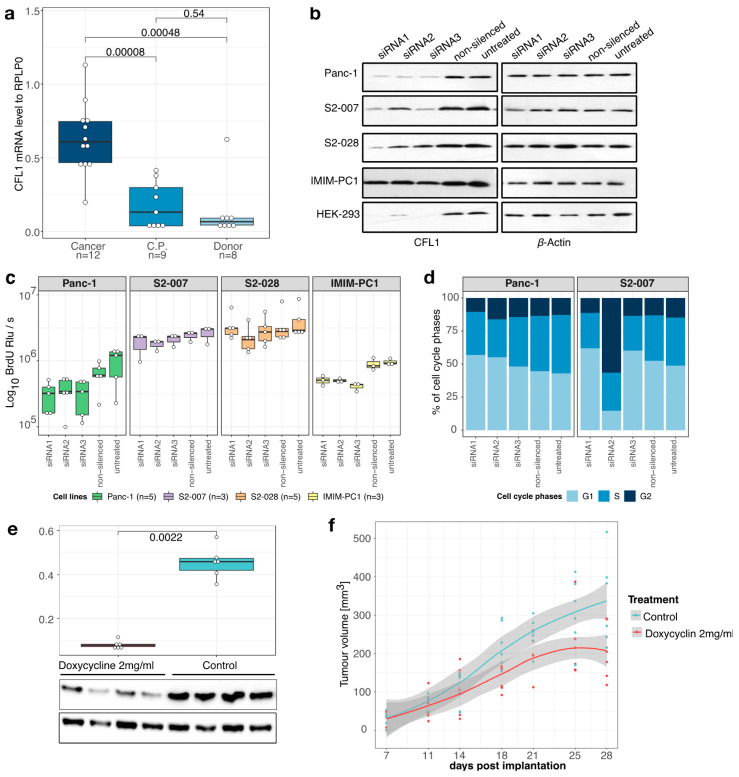
CFL1 in pancreatic cancer. (**a**) Quantitative real-time polymerase chain reaction (qRT-PCR) analyses showed overexpression of CFL1 mRNA in tissue samples of pancreatic ductal adenocarcinoma (Cancer, *n* = 12) compared to chronic pancreatitis (C.P., *n* = 9) or healthy donor (Donor, *n* = 8) pancreas tissues. (**b**) CFL1 knockdown was performed by three independent siRNAs in four pancreatic cancer cell lines (Panc-1, S2-007, IMIM-PC1, and S2-028) and one control cell line (HEK-293). CFL1 knockdown was confirmed on the protein level (left panel). β-actin was used as an equal loading control (right panel). (**c**) BrdU incorporation assays were performed 48 h after the transfection of three independent siRNAs and relative light units per second (Rlu/s) were measured. The number of samples (*n*) represents biological replicates. (**d**) Panc1 and S2-007 cells were propidium-iodide-stained 48 h post-transfection with CFL1-specific or non-silencing siRNAs and subjected to flow cytometric analysis. Both cell lines treated with siRNA1 or siRNA3 show a decrease of cells in S-phase accompanied by an increase of cells in the G1 phase. S2-007 cells transfected with siRNA2 show a dramatic increase of cells in the G2 phase, consistent with G2/M phase arrest. Results depict the average of three independent experiments. (**e**) S2-007 cells with doxycycline-inducible CFL1 knockdown were injected into nude mice. One-half of the mice (*n* = 6) were treated with doxycycline via drinking water to induce CFL1 repression. After the removal of the tumors RNA and protein were extracted from tissues and analyzed for the CFL1 level. Quantitative RT-PCR as well as western blots confirmed repression of CFL1 in the treatment group. β-Actin served as a loading control and mean values of expression were normalized to RPLP0 housekeeping gene. Statistics were performed using Wilcoxon-test. (**f**) Tumor size measurements revealed a significant decrease in tumor growth in doxycycline-treated animals (*n* = 6) compared to controls (*n* = 6). All mice developed tumors. Displayed are data points with a smoothing line and the corresponding confidence interval. Boxplots depict the median with the first and third quartiles.

**Figure 2 cancers-13-00725-f002:**
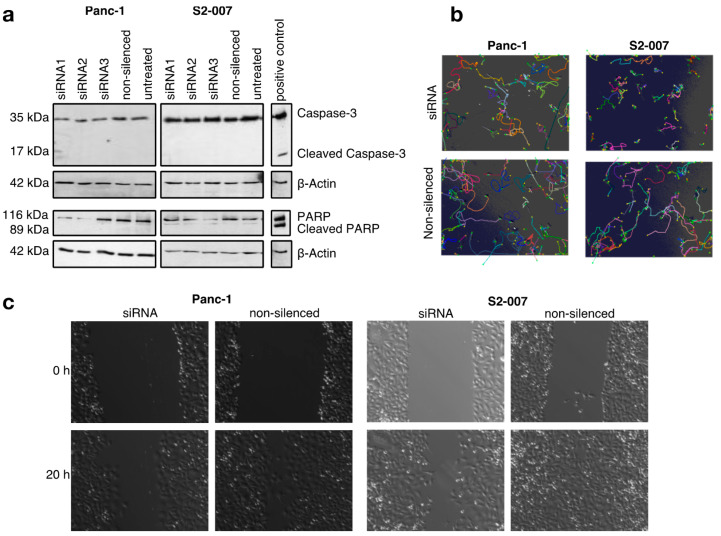
CFL1 knockdown does not induce apoptosis but reduces migration in pancreatic cancer cells (**a**) Western blot analyses with antibodies against full-length and cleaved caspase 3 (upper panels) as well as full length and cleaved poly (ADP-ribose) polymerase (PARP) (lower panels) were performed to check for apoptosis-inducing effects of CFL1 knockdown. UV-treated S2-007 cells served as positive controls for both markers. β-actin was used to ensure equal loading. No cleavage of caspase-3 or PARP was observed following CFL1 knockdown. The results are representatives of three independent experiments. (**b**) Automated time-lapse microscopy was used to determine average cell velocities of individual cells in subconfluent cultures or (**c**) rates of wound closure in confluent cultures of Panc-1 and S2-007 cells. Pictures are representatives of the results.

**Figure 3 cancers-13-00725-f003:**
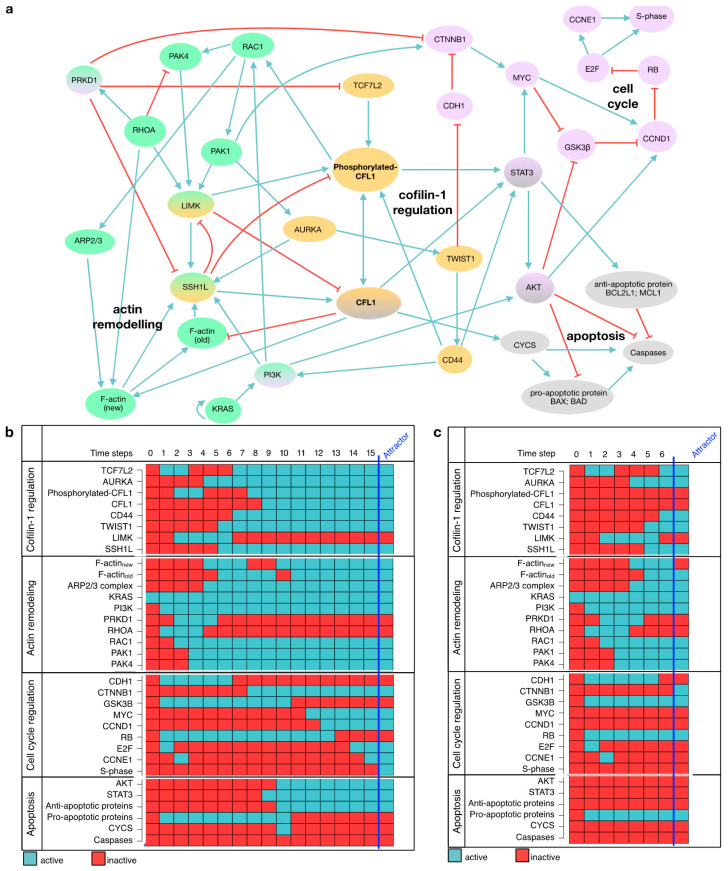
Model of CFL1 in pancreatic cancer. (**a**) Regulatory interactions included in model are depicted in a direct interaction graph. All proteins are shown in colored circles representing the signaling pathway affiliation (blue = actin remodeling, yellow = CFL1 regulation, red = cell cycle, grey = apoptosis). Black arrows depict activating interactions while bar-headed lines represent inhibiting interactions. (**b**) The simulation of a signaling cascade with an unaffected model yields a single state attractor representing pancreatic cancer. This attractor shows active cell cycle components (CCND1, E2F, CCNE1, S-phase) and migratory potential represented by active F-actin (new) and active RAC1 while caspases and thus apoptosis are inactive. (**c**) The simulation of a signaling cascade with an in-silico CFL1 knockout yields another single state attractor representing cell cycle arrest but no induction of apoptosis. Both signaling cascades start from an initial state with only active KRAS as present in 90% of pancreatic cancer patients and proceeds in distinct time steps towards the attractors. Note, that these two attractors are also the only ones that will be obtained by an exhaustive network evaluation. The network components are listed on the left, while the state of each protein is represented by blue (=active) and red (=inactive) rectangles.

**Figure 4 cancers-13-00725-f004:**
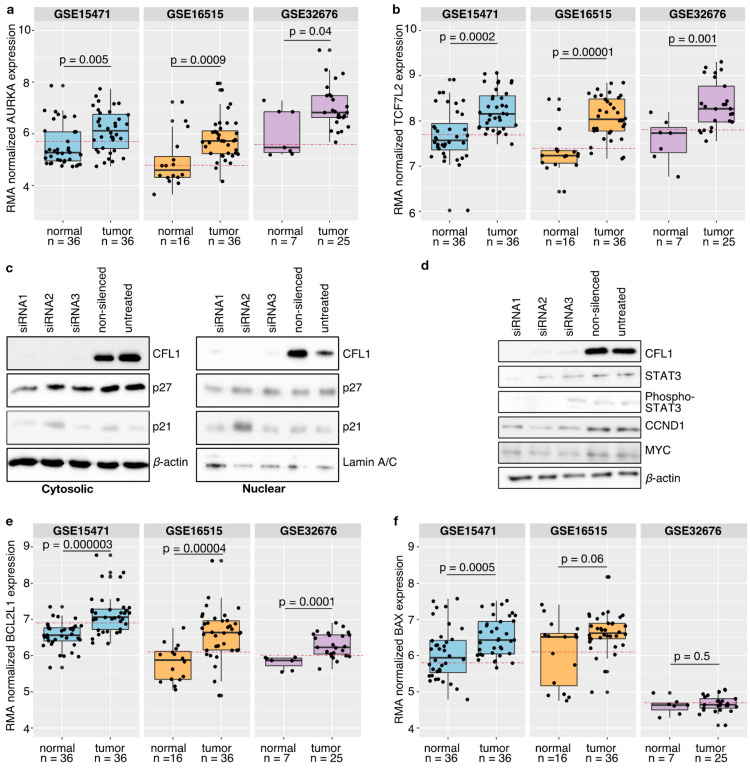
Model validation. (**a**,**b**,**e**,**f**) Three independent publicly available gene expression datasets (GSE15471, GSE16515, GSE32676) from healthy donors and pancreatic cancer tissues were compared with model suggestions. Statistical analyses were performed using the Wilcoxon test and p-values ≤ 0.05 were considered as significant. Only for the dataset GSE15471 containing matched tissues a paired Wilcoxon test was applied. Boxplots depict the median with the first and third quartiles. Binarization of the gene expression by a ROC curve revealed a threshold (red dashed line). Samples above that line are classified as active while samples below that line are considered as inactive. It could be shown that both AURKA (**a**) and TCF7L2 (**b**) are significantly overexpressed in pancreatic cancer patients as suggested by the model. Binarization of the gene expression data revealed that AURKA as well as TCF7L2 are classified as active in pancreatic cancer tissues. (**c**) Western blot analyses with antibodies against CFL1 as well as against cyclin-dependent kinase inhibitors 1A and 1B (p21, p27) were performed to check for involvement in G1 cell cycle arrest. β-actin and Lamin A/C were used to ensure equal loading. Neither cytoplasmic nor nuclear cell fractions of S2-007 cells showed an impact of CFL1 silencing on p21 and p27 protein level thereby excluding the role of these proteins in proliferation regulation. (**d**) Besides, western blot analyses were performed to confirm the model suggested CFL1 regulation on cell cycle regulators in S2-007 cells. CFL1 silencing is accompanied by a reduction of total and active (phosphorylated) STAT3 as well as a decrease of CCND1 and MYC. (**e**,**f**) Similar to the validation of processes leading to CFL1 overexpression and activation, human pancreatic cancer datasets were considered to support the model suggested apoptosis regulation. (**e**) The anti-apoptotic protein BCL2L1 is significantly overexpressed in pancreatic cancer tissues and classified as active. (**f**) Instead, no differences regarding the activity of BAX could be identified. These results further support the model.

**Figure 5 cancers-13-00725-f005:**
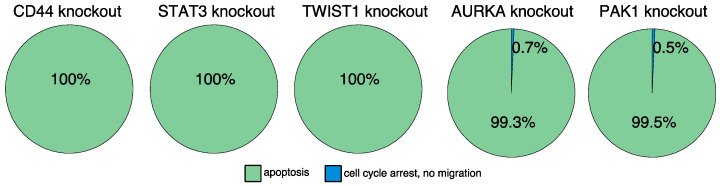
In-silico screening for therapeutic targets. Automated model perturbation screening identified a list of proteins that might induce apoptosis. Displayed is the long-term behavior of the model after introducing the model-suggested interventions. An in-silico knockout of CD44, STAT3, or TWIST1 induce apoptosis. Similar results were obtained for AURKA or PAK1 knockouts. These knockouts mainly induce apoptosis and to a minority cell cycle arrest combined with inhibited migration trades.

**Table 1 cancers-13-00725-t001:** Boolean functions of the CFL1 model. Depicted are the Boolean functions of the analyzed model. Interactions are described by logical connectives AND (∧), OR (∨), and NOT (¬). Linear interactions have been simplified by time delays ((-2) or (-3)).

Node; t + 1	Boolean Function, t	References
TCF7L2	¬PRKD1	PRKD1 inhibits TCF7L2 expression [11].
AURKA	PAK1	PAK1 phosphorylates AURKA [47,48,49].
Phosphorylated-CFL1	CD44 ∨ TCF7L2 ∨ (CFL1 ∧ LIMK ∧ ¬SSH1L)	TCF7L2 activates CFL1 expression [11]. CD44 induces CFL1 expression [50,51]. LIMK inhibits CFL1 [52,53,54,55,56,57]. If both, SSH1L and LIMK are active, CFL1 stays unphosphorylated [58,59,60].
CFL1	(SSH1L ∧ Phosphorylated-CFL1) ∨ (SSH1L ∧ LIMK ∧ Phosphorylated-CFL1)	SSH1L dephosphorylates CFL1 [12,52,53,54,56,57,58,61]. LIMK phosphorylates CFL1 [12,52,53,56,58]. If SSH1L and LIMK are present, LIMK may restore phosphorylation, but the dephosphorylation of SSH1L is more pronounced [58,59,60].
CD44	TWIST1	TWIST1 upregulates CD44 [62,63].
TWIST1	AURKA	AURKA inhibits degradation of TWIST1 [63].
LIMK	(RHOA ∨ PAK1 ∨ PAK4) ∧ ¬SSH1L	LIMK is activated by dephosphorylation of PAK1, PAK4 and ROCK (downstream of RHOA) [48,53,55,59,61,64,65,66]. LIMK and SSH1L can build a complex that effectively dephosphorylates both [59,60].
SSH1L	((F-actin_new_ ∨ F-actin_old_) ∧ ¬PRKD1) ∨ (PI3K ∧ AURKA) ∨ (LIMK ∧ SSH1L)	F-actin enhances SSH1L activity [52,57,59,61]. PRKD1 phosphorylates SSH1L [52,53,61]. In the presence of PI3K, AURKA induces SSH1L expression [54,56,57]. LIMK and SSH1L can build a complex that effectively dephosphorylates both [59,60].
F-actin_new_	(CFL1 ∧ ARP2/3) ∨ (RHOA(-3) ∧ ¬CFL1)	CFL1 and ARP2/3 work in synergy to create new branched actin fibres [58,67,68,69,70,71,72]. RHOA/ROCK/DIA pathway polymerizes F-actin (here RHOA delay) [64,73,74,75].
F-actin_old_	(F-actin_old_ ∧ ¬CFL1) ∨ F-actin_new_	CFL1 severs F-actin [52,65,67] preferring old ADP-F-actin [58,68,76,77]. Newly formed actin fibres are built, prolonged and thus converted into old ones.
ARP2/3	RAC1(-2)	Downstream of RAC1, ARP2/3 is activated by WAVE or WASP (here by a RAC1 delay) [58,64,69].
KRAS	1	Activating KRAS mutations are present in more than 90% of PDAC patients [3,46,78,79]. For this reason, the protein KRAS is assumed to be always active. Therefore, we modeled it as active (1).
PI3K	KRAS ∨ CD44	The PI3K-pathway is one of the main effector pathways downstream of RAS [80]. CD44 receptor binding activates PI3K/AKT pathway [62,81,82].
PRKD1	RHOA	RHOA activates PRKD1 [11,52,53,65].
RHOA	¬PAK4	PAK4 inhibits RHOA [83].
RAC1	PI3K ∨ Phorsphorylated-CFL1(-3)	RAC1 is activated by PI3K [54,80,84]. Phosphorylated CFL1 activates RAC1 via PLD1 and DOCK (here with delay) [85].
PAK1	RAC1	PAK1 is activated by RAC1 [47,48,55,86,87].
PAK4	RAC1	PAK4 is activated by RAC1 [47,48,55].
CDH1	¬TWIST1	TWIST1 inhibits CDH1 expression [63,88,89,90,91].
CTNNB1	PAK1 ∧ ¬PRKD1 ∧ ¬CDH1	PAK1 stabilizes CTNNB1 [48,55,86,87]. CDH1 blocks CTNNB1 entering the nucleus [91,92,93,94]. PRKD1 inhibits CTNNB1 expression [11].
GSK3B	¬AKT	AKT inactivates GSK3B [56,80,95,96,97].
MYC	(¬GSK3B ∧ STAT3) ∨ (CTNNB1 ∧ ¬GSK3B)	STAT3 induces MYC expression [56,98,99,100,101,102]. GSK3B ubiquitinates MYC [103]. GSK3B blocks expression of MYC by CTNNB1 [55,94,104,105].
CCND1	(¬GSK3B ∧ MYC ∧ AKT)	GSK3B destabilizes CCND1 [96,97]. AKT supports the assembly of CCND1 with CDK4/6 [97,106,107,108,109,110]. MYC induces CCND1 expression [95,110,111,112].
RB	¬CCND1	CCND1 inhibits RB [95,96,113].
E2F	¬RB	RB inhibits E2F [95,96,113].
CCNE1	E2F	E2F induces the expression of CCNE1 [95,96].
S-phase	E2F ∧ CCNE1	The synergy of CCNE1 and E2F is responsible for S-phase transition [95,96,103].
AKT	PI3K ∧ STAT3	PI3K activates AKT [54,96,106,110,114]. STAT3 induces expression and activation of AKT [98,115,116,117].
STAT3	(Phosphorylated-CFL1 ∨ CFL1) ∧ CD44	CFL1 regulates amount of STAT3 [45]. CD44 activates STAT3 [56,118].
Anti-apoptotic proteins	STAT3	STAT3 induces expression of BCL2L1 or MCL1 [99,100,101,102,119].
Pro-apoptotic proteins	¬AKT	AKT phosphorylates BAD [119,120,121].
CYCS	Pro-apoptotic proteins ∧ ¬Anti-apoptotic proteins ∧ CFL1	Imbalance between pro- and anti-apoptotic proteins induce release of CYCS by activating BAX. Unphosphorylated CFL1 translocates to the mitochondrion after induction of apoptosis [122,123,124,125] and acts as a carrier for BAX [126].
Caspases	CYCS ∧ ¬AKT	Released CYCS forms an apoptosome further activating caspases signalling [119,120,122,124,127,128,129]. AKT phosphorylates caspase-9 [119,120,121].

Abbreviations: ∧ = and; ∨ = or; ¬ = not; (-2) = time delay of two time steps; (-3) = time delay of three time steps; ADP = Adenosine diphosphate; AKT = protein kinase B; ARP2/3 = actin-related protein-2/3 complex; AURKA = aurora kinase A; Anti-apoptotic proteins = [pooled BAD (=Bcl2-antagonist of cell death), BAX (=Bcl-2-associated X protein), BAK (=Bcl2-anatgonist/killer)]; APAF-1 = apoptotic peptidase activating factor 1; Caspases = pooled caspase-3 and caspase-9 with their corresponding pro-caspases; CCND1 = Cyclin D1; CCNE1 = Cyclin E1; CDH1 = E-cadherin; CD44 = CD44 antigen; CDK4/6 = cyclin dependent kinase 4/6; CFL1 = cofilin-1; CTNNB1 = β-catenin; CYCS = Cytochrome C; DIA = diaphanous related formin; DOCK = dedictor of cytokinesis; E2F = E2F transcription factor; GSK3B = glycogen synthase kinase 3β; KRAS = Kirsten rat sarcoma oncogene; LIMK = LIM domain kinase; MYC = MYC proto-oncogene; PAK1 = p21 activated kinase 1; PAK4 = p21 activated kinase 4; RAC1 = ras-related botulinum toxin substrate 1; RB = retinoblastoma protein; RHOA = ras homolog family member A; PI3K = phosphoinositide-3-kinase; PLD1 = phospholipase D1; PRDK1 = protein kinase D1; Pro-apoptotic proteins = [BCL2L1 (=Bcl2-like 1), MCL1 (=myeloid leukemia cell differentiation protein)]; RAS = rat sarcoma; ROCK = Rho-associated protein kinase; SSH1L = slingshot protein phosphatase 1; STAT3 = signal transducer and activator of transcription 3; TCF7L2 = transcription factor 7-like 2; TWIST1 = twist family BHLH transcription factor 1, WASP = Wiskott-Adrich syndrome protein; WAVE = Verpolin-homologous protein.

**Table 2 cancers-13-00725-t002:** Summary of model validation. The phenotypical behavior of the model is compared to gene expression datasets, laboratory experiments, and previous literature.

Associated Behavior	Model-Based Phenotypical Description (Attractor)	Validation	
		Literature	Wet Lab/Dataset Analyses
Overexpression of active CFL1	PRKD1 (inactive)	[130,131]	
	TCF7L2 (active)		GSE15471, GSE16515, GSE32676 (see Figure 4b)
	AURKA (active)		GSE15471, GSE16515, GSE32676 (see Figure 4a)
	SSH1L (active)	[61]	
Invasion	KRAS (active)	[3,46,78,79]	
	RAC1 (active)	[132]	
	ARP 2/3 (active)	[133,134]	
	F-actin_new_ (active)	[135,136]	Time lapse Figure 2b
Proliferation	STAT3 (active)		Western blot Figure 4d
	AKT (active)	[137,138]	
	MYC (active)		Western blot Figure 4d
	CCND1 (active)		Western blot Figure 4d
Survival	AKT (active)	[137,138]	
	STAT3 (active)		Western blot Figure 4d
	Anti-apoptotic proteins (active)		GSE15471, GSE16515, GSE32676 (see Figure 4e)
	Caspases (inactive)		Western blot Figure 2a

## Data Availability

All data supporting this research are included in the published article or are included in its Appendix A.

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
