# Peer review of "Unraveling the Molecular Tumor-Promoting Regulation of Cofilin-1 in Pancreatic Cancer"

_cancers, 2021, doi:10.3390/cancers13040725_

Round 1

Reviewer 1 Report

The manuscript submitted by Werle and collaborators uncovers the mechanistic modelling of cofilin-1 (CFL1) in pancreatic cancer. This manuscript is divided in two parts. In the first part, the authors charaterized PDAC cell lines that were transfected with siRNA or ShRNA targeting CFL1 and evaluate the impact on cell behaviour and cell signalling. In the later part, the authors built a Boolean model used to predict involvement of CFL1-related partner in proliferation and cell death in order to identify new therapeutic targets.

This manuscript is well written but is really confusing about how is built the mechanistic model. Moreover, the siRNA study suffers of several flaws.

Major points:

  1. My major concern is that it is relatively unclear how the mechanistic model is constructed. It is supposed to be built from experimental observations from the literature, possibly CFL1-KD models but there not information about which one. Are they PDAC KD cells? How were extracted the described regulatory interactions? This urgently needs to be clarified.
  2. The overall mechanistic model result section is extremely confusing. There is a significant mixed part of discussion within the result section. It is sometimes difficult to know which part comes from original result and which part is a comparison with the literature. For example: is there any experimental evidence regarding ARP2/3 or mitochondrial CFL1 in this manuscript.
  3. There is no statistical analysis in Figure S2c and Figure S3. On many occurrences, differences between untreated/non silenced cells and siRNA cells are not convincing. Regarding colony formation, only SiRNA2 condition showed a decrease in S2-007 cells. Similarly, results obtained with Panc1 cells are not convincing in Figure S3 (velocity and wound healing). Notably, siRNA1 has no obvious effect. Only siRNA2 in Panc1 and siRNA2/3 in S2007 show a trend in FigureS3b. non-silenced condition induces an increase of velocity in Figure S3a.
  4. Figure 1d: siRNA2 showed a dramatic effect on cell cycle (flow cytometry) but not the other conditions as the effect is opposite of what is observed with siRNA1 and 3 in S2-007. please explain or discuss
  5. Analysis of GSE15471 should be corrected as there are only 36 individuals analyzed. Not 39 as stated. This dataset contains 3 patients with 2 replicate samples (N30162, N40728 and N41027). Moreover, there are paired samples. Therefore, paired statistical test should be used.
  6. Lane 80, the authors used siRNA from Ambion and control siRNA from thermofisher. They should use siRNA control from the same supplier.
  7. Why the xenograft control group does not receive sucrose 2% water as the experimental group?
  8. The overall quality of figures is of a relatively poor resolution and appears mostly blurry (and small, hard to read…)

Minor points:

  1. Lane 23 “suggested in this article suggesting…” should be rephrased
  2. Lane 91: 2x106 cells should be corrected as 2x10e6
  3. Lane 143 and lane 151 “5000 to 7500 cells”, lane 167 “20000 to 40000 cells”: why do the authors use a range of cell number rather than an exact number?
  4. Lane 433: what is a type2 cell?

Author Response

Reviewer 1

The manuscript submitted by Werle and collaborators uncovers the mechanistic modelling of cofilin-1 (CFL1) in pancreatic cancer. This manuscript is divided in two parts. In the first part, the authors charaterized PDAC cell lines that were transfected with siRNA or ShRNA targeting CFL1 and evaluate the impact on cell behaviour and cell signalling. In the later part, the authors built a Boolean model used to predict involvement of CFL1-related partner in proliferation and cell death in order to identify new therapeutic targets.

This manuscript is well written but is really confusing about how is built the mechanistic model. Moreover, the siRNA study suffers of several flaws.

Major points:

Comment 1:

  • My major concern is that it is relatively unclear how the mechanistic model is constructed. It is supposed to be built from experimental observations from the literature, possibly CFL1-KD models but there not information about which one. Are they PDAC KD cells? How were extracted the described regulatory interactions? This urgently needs to be clarified.

Response 1:

Thank you for this comment. We constructed our model based on extensive literature search on PDAC tumors and CLF1 published data. Pathways that are affected by CFL1 were identified by our initial deep characterization of CFL1 in pancreatic cancer cells and a general literature study about CFL1 in PDAC. The model includes various molecular data coming from both in vitro and in vivo set ups, and, when available, human studies. Nevertheless, we agree that information on model construction was not detailed enough. Therefore, we enlarged the procedural description of the modeling approach in our methods section. Moreover, we moved the table with the Boolean functions from the supplement to the main text and enlarged it by detailed descriptions of all logical functions and interactions endowed with literature references. We truly believe that this comment helped to enhance the quality of our work and its reproducibility.

Comment 2:

  • The overall mechanistic model result section is extremely confusing. There is a significant mixed part of discussion within the result section. It is sometimes difficult to know which part comes from original result and which part is a comparison with the literature. For example: is there any experimental evidence regarding ARP2/3 or mitochondrial CFL1 in this manuscript.

Response 2:

Thank you for raising this issue. We acknowledge that the present work combines a large amount of information integrating different levels of analyses, which therefore deserved a larger space. The present work consists of model-based mechanistic predictions which were validated based on new laboratory experiments and human dataset analyses. While the prediction of mechanisms come solely from the network, its validation can be done by comparison of long-term behaviors (final activities of the proteins). To clarify this, we restructured our results section to state clearly which results (pathway predictions) are coming from the established model, and which validation was applied to confirm these predictions. For this purpose, we either performed new laboratory experiments or compared our model-based behavior with previous studies from literature. Finally, for clarification, we included a new table (Table 2) that summarizes the analysis on the dynamic behavior. Here, it is possible to visually observe which results are provided by the model, and how they were validated either through literature comparison or with new experiments.

Comment 3 :

  • There is no statistical analysis in Figure S2c and Figure S3. On many occurrences, differences between untreated/non silenced cells and siRNA cells are not convincing. Regarding colony formation, only SiRNA2 condition showed a decrease in S2-007 cells. Similarly, results obtained with Panc1 cells are not convincing in Figure S3 (velocity and wound healing). Notably, siRNA1 has no obvious effect. Only siRNA2 in Panc1 and siRNA2/3 in S2007 show a trend in FigureS3b. non-silenced condition induces an increase of velocity in Figure S3a.

Response 3:

Thank you for your comment. We are aware that we did not do inferential statistics for these figures, as we would consider this an overinterpretation on n=3 with multiple comparisions.

Nevertheless, we performed the t-test on our data. Results of the t-test performed on colony formation, wound healing and velocity of both Panc-1 and S2-007 cell lines are reported below. For simplicity we highlighted the statistically significant results.

Still, we contend that these results may be biased by our limited set of samples, and therefore an over interpretation of our current available data should be avoided. On these grounds, we prefer to show our results in the supplement showing trends and tendencies. Given that we focused mainly on unraveling unknown mechanisms involving CLF1 in the regulation of cell cycle and survival, we thought to leave these results on invasion open to further experimental validation. In fact, CLF1 is primarily known for its role in regulating cell motility. Nevertheless, we want to point out that each data point of the cell velocity analyses (Figure S3a) is the average of the cell velocity of multiple cells on one plate. Moreover, a previous study (e.g. Yamauchi et al. 2017) have investigated cell velocity in pancreatic cancer; however, they applied gradients to attract cells. Hence, it might be that this measure is not capturing overall differences between samples in light of a potential heterogeneity of velocities when cells are not equally attracted to a source. Still, we consider these results relevant for the community in the perspective of future studies on tracking of cell motility through live microscopy and cell velocities.

The reviewer is also correct in pointing out that effects were not uniform across all siRNAs and all cell lines. In our view, this strongly underscores the fact that siRNA- or shRNA-mediated knockdown studies are often subject to off-target effects. In order to generate reliable functional data, it is thus of paramount importance to use several interfering RNAs and test different cell lines in order to avoid bot false positive as well as false negative results.

Colonies Panc-1 (Figure S2c)

group1             group2            p p.adj              p.signif            method

siRNA1            siRNA2            0.670               ns                    T-test

siRNA1            siRNA3            0.907               ns                    T-test

siRNA1            non-silenced   0.153               ns                    T-test

siRNA1            untreated         0.0494             *                       T-test

siRNA2            siRNA3            0.561               ns                    T-test

siRNA2            non-silenced   0.223               ns                    T-test

siRNA2            untreated         0.0672             ns                    T-test

siRNA3            non-silenced   0.136               ns                    T-test

siRNA3            untreated         0.0479             *                       T-test

non-silenced   untreated         0.367               ns                    T-test

Colonies S2-007 (Figure S2c)

group1             group2             p p.adj             p.signif            method

siRNA1             siRNA2           0.286               ns                    T-test

siRNA1             siRNA3           0.624               ns                    T-test

siRNA1            non-silenced   0.686               ns                    T-test

siRNA1            untreated         0.596               ns                    T-test

siRNA2             siRNA3           0.155               ns                    T-test

siRNA2             non-silenced 0.157               ns                    T-test

siRNA2            untreated         0.125               ns                    T-test

siRNA3             non-silenced 0.334               ns                    T-test

siRNA3            untreated         0.262               ns                    T-test

non-silenced   untreated         0.898               ns                    T-test

Velocity Panc-1 (Fig. S3a)

 group1             group2                       p p.adj             p.signif             method

siRNA1            siRNA2                        0.312               ns                    T-test

siRNA1            siRNA3                        0.0650             ns                    T-test

siRNA1            non-silenced               0.354               ns                    T-test

siRNA1            untreated                     0.352               ns                    T-test

siRNA2            siRNA3                        0.991               ns                    T-test

siRNA2            non-silenced               0.197               ns                    T-test

siRNA2            untreated                     0.193               ns                    T-test

siRNA3            non-silenced               0.00490           **                     T-test

siRNA3            untreated                     0.00834           **                     T-test

non-silenced   untreated                     0.962               ns                    T-test

Velocity S2-007 (Fig. S3a)

group1             group2                         p p.adj             p.signif             method

siRNA1            siRNA2                        0.421               ns                    T-test

siRNA1            siRNA3                        0.869               ns                    T-test

siRNA1            non-silenced               0.0664             ns                    T-test

siRNA1            untreated                     0.321               ns                    T-test

siRNA2            siRNA3                        0.494               ns                    T-test

siRNA2            non-silenced               0.0352             *                       T-test

siRNA2            untreated                     0.635               ns                    T-test

siRNA3            non-silenced               0.156               ns                    T-test

siRNA3            untreated                     0.370               ns                    T-test

non-silenced   untreated                     0.310               ns                    T-test

Wound Panc-1 (Fig. S3b)

group1             group2                         p p.adj             p.signif             method

siRNA1            siRNA2                        0.0377               *                    T-test

siRNA1            siRNA3                        0.400                 ns                  T-test

siRNA1            non-silenced               0.530                 ns                  T-test

siRNA1            untreated                     0.396                 ns                  T-test

siRNA2            siRNA3                        0.0362               *                    T-test

siRNA2            non-silenced               0.122                 ns                  T-test

siRNA2            untreated                     0.0924               ns                  T-test

siRNA3            non-silenced               0.928                 ns                  T-test

siRNA3            untreated                     0.758                 ns                  T-test

non-silenced   untreated                     0.862                 ns                  T-test

Wound S2-007 (Fig. S3b)

 group1            group2                         p p.adj             p.signif             method

siRNA1            siRNA2                        0.369                  ns                 T-test

siRNA1            siRNA3                        0.320                  ns                 T-test

siRNA1            non-silenced               0.703                  ns                 T-test

siRNA1            untreated                     0.686                  ns                 T-test

siRNA2            siRNA3                        0.928                  ns                 T-test

siRNA2            non-silenced               0.152                  ns                 T-test

siRNA2            untreated                     0.135                  ns                 T-test

siRNA3            non-silenced               0.105                  ns                 T-test

siRNA3            untreated                     0.0861                ns                 T-test

non-silenced   untreated                     0.916                  ns                 T-test

Yamauchi, A., Yamamura, M., Katase, N. et al. Evaluation of pancreatic cancer cell migration with multiple parameters in vitro by using an optical real-time cell mobility assay device. BMC Cancer 17, 234 (2017). https://doi.org/10.1186/s12885-017-3218-4

Comment 4:

  • Figure 1d: siRNA2 showed a dramatic effect on cell cycle (flow cytometry) but not the other conditions as the effect is opposite of what is observed with siRNA1 and 3 in S2-007. please explain or discuss

Response 4:
Please, see also response to comment 3 above: this data point most likely represents a classic off-target effect which is “specific” in that it was highly reproducible, but only with one specific siRNA in one specific cell line. This is now also specifically mentioned in the text

Comment 5:

  • Analysis of GSE15471 should be corrected as there are only 36 individuals analyzed. Not 39 as stated. This dataset contains 3 patients with 2 replicate samples (N30162, N40728 and N41027). Moreover, there are paired samples. Therefore, paired statistical test should be used.

Response 5:

Thank you for pointing this out. We corrected the number of analyzed individuals in Fig. 4 and in our supplement Fig. S1. Concerning the replicated samples, we do not consider a paired test the best solution. Given that a paired test is normally applicable when samples are actually all paired (e.g. before/after treatment), we considered that applying it for 3 replicated data points would finally make the test less conservative.

This would finally lead to a higher likelihood of getting a significant result, but starting from not completely fitting assumptions. Therefore, we proposed another solution to this relevant issue. Since we are dealing with replicate measures that should capture intra-patient variability of expression values, we averaged these data points to better capture this feature. In this way we overcome the issue of having replicate samples in our set, and still consider this data in of our analysis. By doing so, we also re-simulated all these datasets, which are now included in the corresponding figures. Please note that the changes included in the dataset preparation did not affect the results of our analyses nor the final interpretation of the simulations.

Comment 6:

  • Lane 80, the authors used siRNA from Ambion and control siRNA from thermofisher. They should use siRNA control from the same supplier.

Response 6:

Thank you for raising this issue. Actually, this is simply a matter of production dates. Ambion is now part of ThermoFisher. ThermoFisher has confirmed that there are no chemical differences in siRNA composition between batches of control siRNA from different manufacturing dates.

Comment 7:

  • Why the xenograft control group does not receive sucrose 2% water as the experimental group?

Response 7:

Thank you for your question. The sucrose in the drinking water only serves to mask the taste of the doxycycline. This is to ensure that the mice take up enough doxycycline to efficiently induce expression of the shRNAs. The sucrose itself has no influence on tumor growth characteristics as previously established in different studies using the same experimental setup (Kaisha et al. 2017, Buchholz et al. 2003). Nevertheless, we specified this point also in the main text, methods section.

Kaistha BP, Krattenmacher A, Fredebohm J, et al. The deubiquitinating enzyme USP5 promotes pancreatic cancer via modulating cell cycle regulators. Oncotarget. 2017;8(39):66215-66225. Published 2017 Aug 3. doi:10.18632/oncotarget.19882.

Buchholz, Malte, et al. "SERPINE2 (protease nexin I) promotes extracellular matrix production and local invasion of pancreatic tumors in vivo." Cancer Research 63.16 (2003): 4945-4951.

Comment 8:

  • The overall quality of figures is of a relatively poor resolution and appears mostly blurry (and small, hard to read…)

Response 8:

Thank you for raising this issue. Actually, the pictures were provided with respect to the required resolutions. To improve the readability, we reordered the figure panels to allow a bigger scaling in the final draft.

Minor points:

Comment 9:

  • Lane 23 “suggested in this article suggesting…” should be rephrased

Response 9:

We rephrased this sentence and replaced “suggesting” with “indicating”.

Comment 10:

  • Lane 91: 2x106 cells should be corrected as 2x10e6

Response 10:

Thank you for noticing. We corrected the typo as suggested.

Comment 11:

  • Lane 143 and lane 151 “5000 to 7500 cells”, lane 167 “20000 to 40000 cells”: why do the authors use a range of cell number rather than an exact number?

Response 11:

Thank you for pointing this issue out. Due to differences in growth characteristics, the optimal cell number to use in these experiments varies from cell line to cell line. Within one cell line, we always seeded the same number of cells for repeat experiments. Individual numbers for each cell line are now provided in the text.

Comment 12:

  • Lane 433: what is a type2 cell?

Response 12:

Thank for raising this question. The definition of type I or II cells refers to their susceptibility to apoptosis induction (see e.g. Hamacher et al. 2008). While type I cells only require death cell receptors to induce apoptosis, type II cells further need additional mitochondrial signal to trigger cell death. We specified this difference in the main.

Hamacher, R., Schmid, R.M., Saur, D. et al. Apoptotic pathways in pancreatic ductal adenocarcinoma. Mol Cancer 7, 64 (2008). https://doi.org/10.1186/1476-4598-7-64

Reviewer 2 Report

Summary

The authors describe how CFL-1 is a potential target in pancreatic cancer as identified through the literature. They then validated this finding through siRNA knockdown studies. Cell with CFL-1 knockdown showed reduced growth in vivo, reduced migration in vitro and a reduction of apoptosis induction.

Using systems biology/bioinformatic methods the authors exploded other potential pathways containing CFL-1 that may be involved in apoptosis, migration and proliferation consisting of 33 addition nodes or genes.  The authors then used a combination of bioinformatics and literature to validate these interactions. From this validation, the authors suggest CD44 as a particularly attractive target in in pancreatic cancer.

Overall comments

While overall the paper deserves to be published, significant work needs to be done on two main areas.

  • The writing style makes large sections of the paper incomprehensible as the terms and phrases used are ambiguous in meaning in the context used. I would suggest that the authors rewrite or seek assistance in rewriting sections of the paper.
  • The methods section need further clarification and detail.
  • The authors suggest CD44 as a potential pancreatic cancer target and state that CD44 targeting in CFL-1 high pancreatic cancer should be exploded. However, the authors should show in vitro or in vivo results of the inhibition of CD44 using pharmacological inhibitors in CFL-1 high pancreatic cancer cell lines.
  • The authors used 4 cell lines in their work. Panc-1 is the only commercially available line. S2-028, S2-007 and IMIM-PC1 were obtained through collaborations. S2-028 and S2-007 are not primary pancreatic cancer cell lines but derived from liver metastasis. The origins of IMIM-PC1 is unclear. The authors should identify why these cell lines were chosen and should give information in the text as to the nature of the cell lines.
  • Also, given the particular focus placed on the inclusion of the metastatic cell lines it must be questioned if the results here within are relevant to primary pancreatic cancer.
  • No mention is given in the text as to the ethical guidelines or processes followed for the in vivo study.

Specific Comments

Methods

  • Panc-1 cells are an established cell line, known throughout the research community however the other cell lines described are not as well-known and therefore their properties should be described (S2-007, S2-028, IMIM-PC1). That S2-007 and S2-028 are derived from liver metastasis is stated in a figure legend on one of the supplemental figures, when it should be stated when the cells are introduced initially.
  • Method section 2.4-line 91 typographical error.
  • Line 91 - what cells were implanted.
  • Method section 2.6 the isolation of proteins in this manner will result in protein degradation while the samples are being centrifuged at room temperature, standard protocol for protein isolation from adherence cells would involve the uses of lysis buffer and cell scraping to remove the cells from the adherent surface. Cell centrifuge is such a high centrifugation speed would result in cell shear degradation.

Additionally, no information is provided for the transfer or running of the gels for western blotting.

  • Throughout the manuscript all centrifugation speeds should be given as g not rpm
  • Method section 2.7 and 2.8 - if the different cell numbers for seeding density for the assays are relevant to different cell lines than the seeding densities should be given as per cell line so that other authors may repeat these assays
  • Wound scratch method – describe the media composition for this assay – was complete media used? Or were cells tested without FCS present?
  • Method 2.11 for the cell to achieve the confluency were cells seeded at the same seeding density and allowed to achieve confluency over different timeline or were cells seeded at a different seeding densities and allowed to grow for the same timeline to reach the same confluency?
  • Line 198 besides is not the correct term to use here

Results

  • “As a first step in the depth-characterization of CFL1 in pancreatic cancer, we analyzed its expression in primary human pancreatic cancer and control tissues” -State from how many samples in the text.
  • Fig1b, shows western blot of CFL-1 with b-actin as a loading control. Densitometry or some quantitative normalization to the loading control should be completed.
  • The control cell line included, HEK -293, is a normal kidney cell line. Rationale for including this cell line should be stated – is it simply as a normal CFL-1 expressing cell line? Would normal pancreatic cells be a better control?
  • Figure 1c & S2a – please clarify if n= number of biological reps.
  • Fig 1d, missing legend for colours
  • Fig 1f – how many mice were implanted per arm and did all mice grow tumours?
  • Doxycycline addition to drinking water can result in mice choosing not to drink, or to drink less. What parameters were in place to ensure that mice drank comparable quantities of water? Was sucrose added to both water bottles? Crawford et al, PMID 17496224, showed that even with the addition of sucrose to drinking water, animals were dehydrated. Please describe how this was avoided.
  • Fig 1e – “After the explantation of the tumors RNA and protein were extracted from tissues and analyzed for the CFL1 level” Removal or resection would be a more appropriate word for the removal of the tumour from the mouse. Explantation suggests the authors cultured the tumours after in vivo growth. If that is the case, the manuscript needs to be adjusted to reflect that.
  • Figure S7 – Is this protein or gene expression? It is unclear from the image and text which the authors intend this to be. Also clarify the nomenclature for the samples – mouse VR, HR, VL etc means nothing without clarification.
  • Section 3.3, line 284 should read “studies is bladder and vulvar squamous carcinoma show …” as two studies are not “several”.
  • 5 should be rewritten. The text and the communication of the process and findings is convoluted and needs to be clearer. A lot of valuable information is given, particular in the time step graphs, that is not explained or clarified.
  • Line 322/323 “stable state” in this context could mean many things and the authors meanings aren’t clear. Similarly – figure legend for Fig 3 – what is a “single state attractor”.
  • Line 318 cross reference with the relevant section
  • Fig 3 legend contains valuable results date that should be in the results section and not only in the figure legend.
  • Figure 3a – is it possible to change the colours of the arrow line and bar headed lines to make the connections and relationships clearer?
  • Figure S10, the text is not very clear when the image is printed.
  • For the STAT3/CLF-1 work (fig 4d), why was the S2-007 cell line chosen? Given that this in not a primary pancreatic cancer cell line is this the best model to use for this validation?
  • Supplementary figures should be listed in order as they appear in the text.

Author Response

Reviewer 2

Summary

The authors describe how CFL-1 is a potential target in pancreatic cancer as identified through the literature. They then validated this finding through siRNA knockdown studies. Cell with CFL-1 knockdown showed reduced growth in vivo, reduced migration in vitro and a reduction of apoptosis induction.

Using systems biology/bioinformatic methods the authors exploded other potential pathways containing CFL-1 that may be involved in apoptosis, migration and proliferation consisting of 33 addition nodes or genes.  The authors then used a combination of bioinformatics and literature to validate these interactions. From this validation, the authors suggest CD44 as a particularly attractive target in in pancreatic cancer.

Overall comments

While overall the paper deserves to be published, significant work needs to be done on two main areas.

Comment 1:

The writing style makes large sections of the paper incomprehensible as the terms and phrases used are ambiguous in meaning in the context used. I would suggest that the authors rewrite or seek assistance in rewriting sections of the paper.

Response 1:

Thank you for your comment. We restructured our text to make it clearer to the reader, especially in our results part. Here, we clarified what comes from our original work and what is instead compared/inferred from literature. Furthermore, to make our in-silico simulations more clear we now provided a new table (Table 2) that summarizes the main points of our results.

Comment 2:

The methods section need further clarification and detail.

Response 2:

Thank you for raising this issue. We enlarged the method section to clarify the procedure of performed experiments.

Comment 3:

The authors suggest CD44 as a potential pancreatic cancer target and state that CD44 targeting in CFL-1 high pancreatic cancer should be exploded. However, the authors should show in vitro or in vivo results of the inhibition of CD44 using pharmacological inhibitors in CFL-1 high pancreatic cancer cell lines.

Response 3:

Thank you for this suggestion. We actually discussed previous pre-clinical work on CD44 inhibition in our manuscript in the discussion section. CD44 has been characterized in pancreatic cancer cell lines and mouse models with respect to its connection to tumor proliferation and invasiveness. Moreover, also in vivo models show more susceptibility to pharmacological targets when CD44 is at low concentrations. Interestingly, in one of these studies also Panc-1 cells are investigated, which we now showed to have high CLF-1 expression. On these grounds, even if the previous work on CD44 needs still to be further enlarged, we do not feel that for our current work further experiments would be needed. We would also like to point out that our in-silico approach predicts and supports the evaluation of different intervention targets in CLF-1 high PDACs. This not to be underestimated given the potentiality of in-silico approaches to reduce the experimental effort. Our suggestions are supported by initial results from other groups (see Nambiar et al. 2013, Li et al. 2015, Zhao et al. 2016, Yan et al. 2016).

Nevertheless, we agree with the reviewer that the further details should be given on this point. Therefore, we enlarged our references and discussion on this issue.

Comment 4:

The authors used 4 cell lines in their work. Panc-1 is the only commercially available line. S2-028, S2-007 and IMIM-PC1 were obtained through collaborations. S2-028 and S2-007 are not primary pancreatic cancer cell lines but derived from liver metastasis. The origins of IMIM-PC1 is unclear. The authors should identify why these cell lines were chosen and should give information in the text as to the nature of the cell lines.

Response 4:

Although several of the cell lines were indeed originally derived from liver metastases of pancreatic tumors (mostly due to practical considerations concerning clinical workup of PDAC patients), this does not invalidate their use as in vitro models of PDAC, since many studies have established that metastases share the phenotypic traits of the primary tumor from which they derive (see e.g. Zhong et al. 2017, Mueller et al. 2018).

We chose the cell lines because they represent a spectrum of different grades of differentiation and invasiveness, with the goal of avoiding to falsely identify functional effects that are in reality only artifacts present in a single cell line.

The information on the cell lines has now been extended to more clearly identify their origin.

Zhong, Yi, et al. "Mutant p53 together with TGFβ signaling influence organ-specific hematogenous colonization patterns of pancreatic cancer." Clinical Cancer Research 23.6 (2017): 1607-1620.

Mueller, Sebastian, et al. "Evolutionary routes and KRAS dosage define pancreatic cancer phenotypes." Nature 554.7690 (2018): 62-68.

Comment 5:

Also, given the particular focus placed on the inclusion of the metastatic cell lines it must be questioned if the results here within are relevant to primary pancreatic cancer.

Response 5:

According to our reply to comment 4, we feel confident that results obtained with these cell lines also reflect molecular principles that are operative in cell lines derived from primary tumors, since both share a similar origin.

Comment 6:

No mention is given in the text as to the ethical guidelines or processes followed for the in vivo study.

Response 6:

Thank you for pointing out this important issue. Animal experiments were approved by the relevant Ethics Committee at the Regierungspräsidium Giessen, Germany (ethic approval number V54 19c 20 15 (1) MR 20/11 Nr. 50/2011). We provided now this statement in the “Institutional Review Board Statement” as well as in the method section below the in vivo experiment.

Specific Comments

Methods

Comment 7:

Panc-1 cells are an established cell line, known throughout the research community however the other cell lines described are not as well-known and therefore their properties should be described (S2-007, S2-028, IMIM-PC1). That S2-007 and S2-028 are derived from liver metastasis is stated in a figure legend on one of the supplemental figures, when it should be stated when the cells are introduced initially.

Response 7:

Thank you for your comment. As mentioned in the reply to the comment 4, this information is now provided upfront in the Methods section.

Comment 8:

Method section 2.4-line 91 typographical error.

Response 8:

Thank you for pointing out the typo. This error has now been corrected as  2 x 106 cells.

Comment 9:

Line 91 - what cells were implanted.

Response 9:

The cell line used in these experiments were S2-007 cells. This information has now been added.

Comment 10:

Method section 2.6 the isolation of proteins in this manner will result in protein degradation while the samples are being centrifuged at room temperature, standard protocol for protein isolation from adherence cells would involve the uses of lysis buffer and cell scraping to remove the cells from the adherent surface. Cell centrifuge is such a high centrifugation speed would result in cell shear degradation.

Response 10:

This is an error in the manuscript for which we would like to apologize. Cells were always centrifuged at 4°C to avoid the degradation that you correctly mentioned. This has now been corrected in the manuscript.

Comment 11:

Additionally, no information is provided for the transfer or running of the gels for western blotting.

Response 11:

We apologize for this inconvenience. This information is now provided in the manuscript.

Comment 12:

Throughout the manuscript all centrifugation speeds should be given as g not rpm

Response 12:

Thank you for raising this point. We agree with the reviewer that rpm is dependent on the specific centrifuge and thereby might affect the reproducibility of the experiments. We now provided g values for all our centrifugation steps.

Comment 13:

Method section 2.7 and 2.8 - if the different cell numbers for seeding density for the assays are relevant to different cell lines than the seeding densities should be given as per cell line so that other authors may repeat these assays

Response 13:

Thank you for pointing this out. Individual numbers for the different cell lines are now provided.

Comment 14:

Wound scratch method – describe the media composition for this assay – was complete media used? Or were cells tested without FCS present?

Response 14:

These assays were conducted with complete media. This is now clarified in the Methods section.

Comment 15:

Method 2.11 for the cell to achieve the confluency were cells seeded at the same seeding density and allowed to achieve confluency over different timeline or were cells seeded at a different seeding densities and allowed to grow for the same timeline to reach the same confluency?

Response 15:

Cells were seeded at different densities to achieve confluence within similar time lines.

Comment 16

Line 198 besides is not the correct term to use here

Response 16:

Thank you for your comment. We restructured the whole paragraph and corrected this issue.

Results

Comment 17:

As a first step in the depth-characterization of CFL1 in pancreatic cancer, we analyzed its expression in primary human pancreatic cancer and control tissues” -State from how many samples in the text.

Response 17:

Thank you for pointing this out. This information has now been included in the main text.

Comment 18:

Fig1b, shows western blot of CFL-1 with b-actin as a loading control. Densitometry or some quantitative normalization to the loading control should be completed.

Response 18:

Thank you for your suggestion. The densitometric analysis is now included in the supplement (Fig. S2b) and is referred to this figure in the main text. Besides, we extended our methods to include the procedure we followed for this analysis. 

Comment 19:

The control cell line included, HEK -293, is a normal kidney cell line. Rationale for including this cell line should be stated – is it simply as a normal CFL-1 expressing cell line? Would normal pancreatic cells be a better control?

Response 19:

Yes, HEK-293 were used as a well-characterized normal (non-tumor) CFL1-expressing control cell line (this is now stated in the text). Normal pancreatic cell lines would not necessarily have been a better alternative, since they would also have to be immortalized to be able to continuously grow in cell culture, and would thus potentially suffer from artifacts of immortalization.

Comment 20:

Figure 1c & S2a – please clarify if n= number of biological reps.

Response 20:

Yes, n= signifies the number of biological replicates. This is now explicitly stated in the figure legends.

Comment 21:

Fig 1d, missing legend for colours

Response 21:

Thank you for pointing this out. We lost this information by mistake and was now added to the figure.

Comment 22:

Fig 1f – how many mice were implanted per arm and did all mice grow tumours?

Response 22:

Yes, all mice developed tumors. The stated number of animals (n=6 per arm, 12 total) is thus identical with the number of animals implanted with tumors. We further specified this point the figure legend and in the methods section.

Comment 23:

Doxycycline addition to drinking water can result in mice choosing not to drink, or to drink less. What parameters were in place to ensure that mice drank comparable quantities of water? Was sucrose added to both water bottles? Crawford et al, PMID 17496224, showed that even with the addition of sucrose to drinking water, animals were dehydrated. Please describe how this was avoided.

Response 23:

Thank you for raising this interesting point. This format of xenograft studies with doxycycline-inducible constructs is well-established in our group; see e.g. Kaistha et al., Oncotarget. (2017) 8:66215-66225; Buchholz et al., Cancer Res. (2003) 63:4945-51. In particular, the parameters described by Crawford et al. (skin elasticity, cyanotic appearance,“thinning” of the skin, and weight loss) are carefully controlled for in all our experiments as part of our general animal husbandry guidelines and are recorded on score sheets for every individual mouse. Strict adherence to this procedure is routinely controlled by the relevant authorities as part of our ethics approval procedures for animal experiments. We can definitely exclude these effects in our experiments; it should be noted in this context that in contrast to Crawford et al., we did not provide drinking water with sucrose to the mice prior to adding doxycycline (which may have made the animals reject the altered taste after they had already become accustomed to the sucrose), and we used 2% sucrose (compared to 1% used by Crawford et al.), which may be more effective in masking the doxycycline.

Kaistha BP, Krattenmacher A, Fredebohm J, et al. The deubiquitinating enzyme USP5 promotes pancreatic cancer via modulating cell cycle regulators. Oncotarget. 2017;8(39):66215-66225. Published 2017 Aug 3. doi:10.18632/oncotarget.19882.

Buchholz, Malte, et al. "SERPINE2 (protease nexin I) promotes extracellular matrix production and local invasion of pancreatic tumors in vivo." Cancer Research 63.16 (2003): 4945-4951.

Comment 24:

Fig 1e – After the explantation of the tumors RNA and protein were extracted from tissues and analyzed for the CFL1 level” Removal or resection would be a more appropriate word for the removal of the tumour from the mouse. Explantation suggests the authors cultured the tumours after in vivo growth. If that is the case, the manuscript needs to be adjusted to reflect that.

Response 24:

As correctly pointed out from the reviewer we meant that the tumor was resected/removed. We have changed the wording in the main accordingly.

Comment 25:

Figure S7 – Is this protein or gene expression? It is unclear from the image and text which the authors intend this to be. Also clarify the nomenclature for the samples – mouse VR, HR, VL etc means nothing without clarification.

Response 25:

This figure represents protein expression, since these are simply the uncropped versions of the western blots shown in Fig. 1e. The VR, HR, etc labels have been removed to avoid confusion, since they only served to identify individual mice and do not add any relevant information to this figure.

Comment 26:

Section 3.3, line 284 should read “studies is bladder and vulvar squamous carcinoma show …” as two studies are not “several”.

Response 26:

According to your comment, we changed the wording accordingly.

Comment 27:

5 should be rewritten. The text and the communication of the process and findings is convoluted and needs to be clearer. A lot of valuable information is given, particular in the time step graphs, that is not explained or clarified.

Response 27:

Thank you for your comment. We restructured and rewrote our results focusing on giving a detailed description of the model construction and simulation. Furthermore, two tables were added regarding these points to support the reader in better understanding the amount of given information. Finally, we enlarged our methods sections on model construction and analysis as a further addition to the general understanding of these results paragraphs.

Comment 28:

Line 322/323 “stable state” in this context could mean many things and the authors meanings aren’t clear. Similarly – figure legend for Fig 3 – what is a “single state attractor”.

Response 28:

We apologize that this notion was not introduced in the manuscript. This information is now stated in our methods section and in the main.

Comment 29:

Line 318 cross reference with the relevant section

Response 29:

Thank you for your comment. We now cross referenced the statement with the relevant previous sections of the results.

Comment 30:

Fig 3 legend contains valuable results date that should be in the results section and not only in the figure legend.

Response 30:

We agree with the reviewer that not enough space was given to the description of the model building process and the dynamic analysis. To extensively cover these points, we restructured and enlarged bot hour results and methods sections.

Comment 31:

Figure 3a – is it possible to change the colours of the arrow line and bar headed lines to make the connections and relationships clearer?

Response 31:

We colored the arrows following the legend of our attractor pattern (Fig. 3 b and c). Accordingly, Fig.3 a colors were adapted to avoid overlapping patterns between nodes and arrows.

Comment 32:

Figure S10, the text is not very clear when the image is printed.

Response 32:

We apologize for the inconvenience. We corrected the quality of the figure to make it visible in the printed version.

Comment 33:

For the STAT3/CLF-1 work (fig 4d), why was the S2-007 cell line chosen? Given that this in not a primary pancreatic cancer cell line is this the best model to use for this validation?

Response 33:

As discussed in the reply to comment 4, we do not expect cells from primary tumors or metastases to fundamentally differ in basic molecular mechanisms. Therefore, we opted to use a cell line with comparatively high CFL1 expression levels and good response to siRNA-mediated knockdown (see Fig. 1b). The results in Fig. 4d seem to confirm that S2-007 cells were an appropriate choice for the validation experiments. Given the interest that the selection of cells lines raised, we included these considerations in the discussion of the manuscript. We thank the reviewer for opening this interesting discussion, that we believe will be of interest also for the scientific community belonging to this research field.

Comment 34

Supplementary figures should be listed in order as they appear in the text.

Response 34:

Thank you for your comment. We have changed the order of the supplementary figures accordingly.

Round 2

Reviewer 1 Report

The authors provided a improved manuscript with much detailed explanation of their algorithm.

I have two remaining minor concern/question

1- what does mean the "1" in the second column on table 1 (last lane of the first page)?

KRAS         1                   Activating KRAS mutations are present in more than 90% of PDAC pa-tients [3,7779].

2- I think my comment #5 regarding GSE15471 was not entirely clear. The authors corrected nicely the small problem with the 3 replicates.
However, regarding my last sentence, I suggested to perform a paired statistical test to compare the 36 normal tissues the paired 36 tumor samples (after removing the 3 replicates). I still believe this would be more appropriate even thought there would be probably no difference as it would remain statistically significative..

This manuscript is fit for publication.

Author Response

We thank the reviewer for the comments. Please find our point-to-point replies below.
The authors provided a improved manuscript with much detailed explanation of their algorithm.
We are glad you appreciate our effort in clarifying our in silico approach. We throughly think the comments during the first round of revision helped to increase the quality of our work.
I have two remaining minor concern/question
Comment 1:
1- what does mean the "1" in the second column on table 1 (last lane of the first page)?
KRAS 1 Activating KRAS mutations are present in more than 90% of
PDAC pa-tients [3,77–79].
Response 1:
Thank you for your comment. It is possible in modeling approaches to represent genetic alterations as constitutive knock-ins or knock-outs. In these cases, the activity of these proteins is set to a constant value and its regulatory Boolean function is not applied. This represents the idea of gain and loss of function mutations where the finial activity is independent from the environment. As you correctly mentioned, this is the case for KRAS mutations. We briefly clarified this regulatory concept in table 1 column KRAS highlighted in red.

Comment 2:
2- I think my comment #5 regarding GSE15471 was not entirely clear. The authors corrected nicely the small problem with the 3 replicates. However, regarding my last sentence, I suggested to perform a paired statistical test to compare the 36 normal tissues the paired 36 tumor samples (after removing the 3 replicates). I still believe this would be more appropriate even thought there would be probably no difference as it would remain statistically significative..
This manuscript is fit for publication.
Response 2:
We thank the reviewer for correctly insisting on this point. Actually we acknowledge that there was a misunderstanding of comment 5. We agree with the reviewer that for this dataset a paired test is suitable given that these samples are obtained from both pancreatic tumors and surrounding healthy tissue at the time of resection. We added the new p-values to figure 4 and mentioned the paired-test for this dataset.
